# Pointwise uncertainty quantification for sparse variational Gaussian process regression with a Brownian motion prior

**Luke Travis**
Department of Mathematics
Imperial College London
luke.travis15@imperial.ac.uk

**Kolyan Ray**
Department of Mathematics
Imperial College London
kolyan.ray@imperial.ac.uk

## Abstract

We study pointwise estimation and uncertainty quantification for a sparse variational Gaussian process method with eigenvector inducing variables. For a rescaled Brownian motion prior, we derive theoretical guarantees and limitations for the frequentist size and coverage of pointwise credible sets. For sufficiently many inducing variables, we precisely characterize the asymptotic frequentist coverage, deducing when credible sets from this variational method are conservative and when overconfident/misleading. We numerically illustrate the applicability of our results and discuss connections with other common Gaussian process priors.

## 1 Introduction

Consider the standard nonparametric regression model, where we observe $n$ training samples $\mathcal{D}_n = \{(x_1, y_1), \ldots, (x_n, y_n)\}$ arising from the model

$$y_i = f(x_i) + \varepsilon_i, \qquad \varepsilon_i \sim^{iid} \mathcal{N}(0, \sigma^2), \qquad (1)$$

where $\sigma^2 > 0$ and the design points $x_i \in \mathcal{X} \subset \mathbb{R}^d$ are either fixed or considered as i.i.d. random variables. Our goal is to predict outputs $y^*$ based on new input features $x^*$, while accounting for the statistical uncertainty arising from the training data. A widely-used Bayesian approach is to endow $f$ with a Gaussian process (GP) prior [25], which is especially popular due to its ability to provide *uncertainty quantification* via posterior credible sets.

While explicit expressions for the posterior distribution are available, a well-known drawback is that these require $O(n^3)$ time and $O(n^2)$ memory complexity, making computation infeasible for large data sizes $n$. To avoid this, there has been extensive research on low-rank GP approximations, where one chooses $m \ll n$ inducing variables to summarize the posterior, thereby reducing computation to $O(nm^2)$ time and $O(nm)$ memory complexity, see the recent review [20].

We consider here the sparse Gaussian process regression (SGPR) approach introduced by Titsias [31], which is widely used in practice (see [1, 9] for implementations) and has been studied in many recent works [13, 21, 5, 6, 38, 28, 32, 22, 23]. While the computational dependence on the number of inducing variables $m$ is relatively well-understood, fewer theoretical results are available on understanding how $m$ affects the quality of statistical inference. In particular, it is crucial to understand how large $m$ needs to be to achieve good statistical uncertainty quantification, ideally close to that of the true computationally expensive posterior. The present work contributes to this research direction by establishing precise theoretical guarantees for pointwise inference using a natural choice of SGPR with eigenvector inducing variables.

A main appeal of Bayesian methods is their ability to perform uncertainty quantification (UQ) via credible sets, often at the 95% level, see Figure 1 for a visualization involving a full GP and two

37th Conference on Neural Information Processing Systems (NeurIPS 2023).

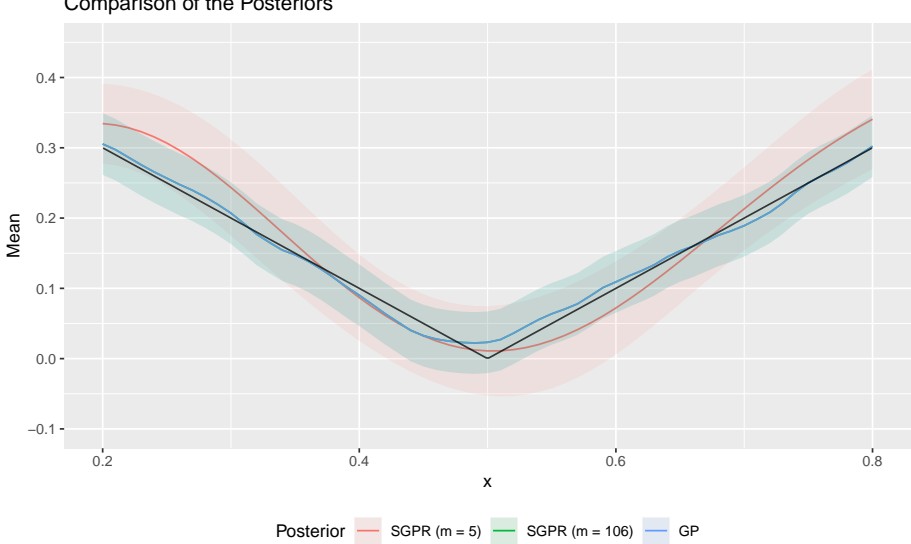

Figure 1: Plot of the (sparse) posterior means with the ribbons representing 95% posterior pointwise credible intervals for the i) SGPR with $m = 5$ inducing variables (red); ii) SGPR with $m = 106$ inducing variables (green); iii) full GP (blue) and iv) the true function (black). Here, $n = 500$ and the covariance kernel is the rescaled Brownian motion with $\gamma = 0.5$.

SGPR approximations. We focus here on the canonical problem of pointwise estimation and UQ for $f(x)$ at a single point $x \in \mathcal{X}$, which corresponds to the practically relevant problem of quantifying the uncertainty of a prediction $f(x)$ based on a new input $x$. We study the frequentist behaviour of the resulting SGPR credible sets, with our main conclusion being that for well-calibrated priors and sufficiently many inducing points $m$, SGPRs can indeed provide reliable, though conservative, uncertainty quantification.

Concretely, we consider a rescaled Brownian motion prior that models Hölder smooth functions [33, 36, 29] and whose pointwise UQ for the corresponding full posterior was studied in [29]. We extend this analysis to the SGPR setting using the eigendecomposition of the sample covariance matrix as inducing variables, i.e. the optimal rank-$m$ posterior approximation. We provide a rigorous theoretical justification for pointwise estimation and UQ using this SGPR, provided $m$ is large enough. Our main theoretical contributions are summarized that in dimension $d = 1$, for an $\alpha$-smooth truth $f_0$, $\gamma$-smooth prior, $\alpha, \gamma \in (0, 1]$, and at least $m \gg n^{\frac{1}{1+2\gamma}(\frac{2+\alpha}{1+\alpha})}$ inducing variables:

1. The SGPR recovers the true value $f_0(x)$ at a point $x$ with convergence rate $n^{-\frac{\min(\alpha,\gamma)}{2\gamma+1}}$. If $\alpha = \gamma$, this gives the minimax optimal rate $n^{-\frac{\alpha}{2\alpha+1}}$.

2. For truths that are smoother than the prior ($\alpha > \gamma$), SGPR credible sets are conservative with their frequentist coverage converging to a value strictly between their nominal level and one, which we characterize.

3. For truths that are rougher than the prior ($\alpha \leq \gamma$), coverage can be arbitrarily bad.

4. It is not necessary for the KL divergence between the SGPR and true posterior to vanish as $n \to \infty$ to get good (though conservative) coverage.

5. We provide a more general small-bias condition under which the coverage of SGPR credible sets is at least that of credible sets from the true posterior.

Our proofs exploit specific properties of Brownian motion to obtain explicit expressions, which allow us to obtain the above precise results. Such expressions seem difficult to obtain for other commons GPs (e.g. Matérn, squared exponential), meaning that existing theoretical guarantees for the Matérn SGPR apply to *enlarged* credible sets [32, 23]. Since our results indicate that pointwise SGPR credible sets are already conservative, such enlargements make the SGPR UQ procedure very

conservative, see Section 4. We provide a complimentary perspective to these works, studying *true* credible sets from a related GP, rather than modified credible sets from the Matérn.

The Brownian motion prior we consider here shares many similarities with the Matérn process, meaning our main qualitative conclusions also apply there, as we demonstrate in simulations. We thus verify the applicability of our limit theorems in numerical simulations, confirming they provide a good guide to the behaviour of SGPRs for finite sample sizes, including those based on Matérn priors. To the best of our knowledge, our work is the first establishing exact theoretical coverage guarantees for SGPR pointwise credible sets, and can be viewed as a guide of what to expect for the other Gaussian processes, see Section 4.

**Related work.** Rigorous theoretical guarantees for Bayesian UQ using the full posterior have received much recent attention. In parametric models, the classic Bernstein–von Mises theorem [35] ensures that regular credible sets are asymptotic confidence sets, justifying Bayesian UQ from a frequentist perspective. However, such a result fails to hold in high- and infinite-dimensional settings [10, 16], such as GPs, where the performance is sensitive to the choice of prior and model. Theoretical guarantees for UQ using full GPs have been established mainly involving $L^2$-type credible sets (e.g. [17, 7, 30]) which, while mathematically more tractable, are less reflective of actual practice than pointwise credible sets. Results for pointwise Bayesian UQ are more difficult to obtain and are correspondingly much rarer [29, 40, 39], see Section 3. From a technical perspective, we perform a careful and novel analysis of the gap between the pointwise behaviour of the true posterior and its SGPR approximation, after which we can invoke ideas from the proofs for the full posterior [29].

While the frequentist asymptotic properties of VB have increasingly been investigated, this is usually in the context of estimation, convergence rates or approximation quality (e.g. [24, 2, 41, 27, 26], including for SGPRs [5, 6, 22]) rather than UQ. Available results on mean-field VB often show that this factoriazable approach provides overconfident UQ by underestimating the posterior variance [3, 4], though there exist more positive results in certain low-dimensional models [37]. Regarding UQ for SGPRs, Nieman et al. [23] investigate the frequentist coverage of global $L^2$-credible sets from certain SGPRs, showing these can have coverage tending to 1, possibly after enlarging them. Vakili et al. [32] similarly show that enlarged pointwise or uniform credible sets can have good coverage in a different bandit-like setting.

**Organization.** In Section 2 we detail the problem setup, including notation and an overview of (sparse variational) Gaussian process regression. Main results on pointwise convergence and UQ for SGPRs based on a Brownian motion prior are found in Section 3, discussion on the connections with other GPs in Section 4, simulations in Section 5 and discussion in Section 6. In the supplement, we provide the proofs of our results (Section A) and additional simulations (Section B).

## 2 Problem setup and sparse variational Gaussian process regression

Recall that we observe $n$ training examples $\mathcal{D}_n = \{(x_1, y_1), \ldots, (x_n, y_n)\}$ from model (1), and write $\mathbf{y} = (y_1, \ldots, y_n)^T$ and $\mathbf{f} = (f(x_1), \ldots, f(x_n))^T$. We denote by $P_f$ the probability distribution of $\mathbf{y}$ from model (1) and by $E_f$ the corresponding expectation. For $\mathbf{u}, \mathbf{v} \in \mathbb{R}^n$ we write $\langle \mathbf{u}, \mathbf{v} \rangle = \sum_{i=1}^n \mathbf{u}_i \mathbf{v}_i$ for the inner-product on $\mathbb{R}^n$ and $\|\mathbf{u}\| = \langle \mathbf{u}, \mathbf{u} \rangle^{1/2}$ for the usual Euclidean norm. Write $C(\mathcal{X})$ for the space of continuous real-valued functions on $\mathcal{X}$. For $\alpha \in (0, 1]$, we further define the class of Hölder smooth functions on $[0, 1]$ by $C^\alpha = C^\alpha[0, 1] = \{f : [0, 1] \to \mathbb{R} : \sup_{x \neq y} \frac{|f(x) - f(y)|}{|x - y|^\alpha} < \infty\}$. Throughout the paper we make the following frequentist assumption:

**Assumption 2.1.** *There is a true $f_0 \in L^2(\mathcal{X})$ generating the data $\mathbf{y} \sim P_{f_0}$ according to* (1).

We differentiate $f$ coming from the Bayesian model with the 'true' generative $f_0$.

### 2.1 Sparse variational Gaussian process regression

In model (1), we assign to $f \sim GP(\nu_0, k)$ a Gaussian process (GP) prior with mean function $\nu_0 : \mathcal{X} \to \mathbb{R}$ and covariance kernel $k : \mathcal{X} \times \mathcal{X} \to \mathbb{R}$. We henceforth take the prior mean $\nu_0 = 0$ for simplicity since this will not alter the spirit of our results, see Remark 3.3 below. The resulting

posterior distribution is again a GP by conjugacy [25], with mean and covariance functions equal to

$$\nu_n(x) = \mathbf{k}_n(x)^T(\mathbf{K}_{nn} + \sigma^2\mathbf{I}_n)^{-1}\mathbf{y},$$
$$k_n(x, x') = k(x, x') - \mathbf{k}_n(x)^T(\mathbf{K}_{nn} + \sigma^2\mathbf{I}_n)^{-1}\mathbf{k}_n(x'), \tag{2}$$

where $\mathbf{k}_n(x) = (k(x, x_1), \dots, k(x, x_n))^T$ and $[\mathbf{K}_{nn}]_{ij} = k(x_i, x_j)$. In particular, the posterior variance at a point $x \in \mathcal{X}$ is $\sigma_n^2(x) = k_n(x, x)$. In Section 3.2 below, we will specifically consider the choice $\mathcal{X} = [0, 1]$ and rescaled Brownian motion prior kernel $k(x, x') = (n+1/2)^{\frac{1-2\gamma}{1+2\gamma}}\min(x, x')$ for $\gamma > 0$.

We consider here the sparse variational Gaussian process (SGPR) approximation using inducing variables proposed by Titsias [31]. The idea is to summarize the posterior via a relatively small number of inducing variables $\mathbf{u} = \{u_1, \dots, u_m\}$, $m \ll n$, which are linear functionals of the prior process $f$. By assigning $\mathbf{u}$ a multivariate normal distribution, one obtains a low-rank variational family. Titsias [31] explicitly computed the minimizer in Kullback Leibler sense between this family of GPs and the full posterior, which yields the SGPR $Q^* = GP(\nu_m, k_m)$ with mean and covariance

$$\nu_m(x) = \mathbf{K}_{xm}(\sigma^2\mathbf{K}_{mm} + \mathbf{K}_{mn}\mathbf{K}_{nm})^{-1}\mathbf{K}_{mn}\mathbf{y}$$
$$k_m(x, x') = k(x, x') - \mathbf{K}_{xm}\mathbf{K}_{mm}^{-1}\mathbf{K}_{mx'} + \mathbf{K}_{xm}(\mathbf{K}_{mm} + \sigma^{-2}\mathbf{K}_{mn}\mathbf{K}_{nm})^{-1}\mathbf{K}_{mx'}, \tag{3}$$

where $\mathbf{K}_{xm} = (\text{cov}(f(x), u_1), \dots, \text{cov}(f(x), u_m)) = \mathbf{K}_{mx}^T$, $[\mathbf{K}_{mm}]_{ij} = \text{cov}(u_i, u_j)$ and we write $[\mathbf{K}_{nm}]_{ij} = \text{cov}(f(x_i), u_j)$. Recalling that $\text{cov}(f(x), f(x')) := k(x, x')$ under the prior, for $u_j$ of the form $u_j = \sum_{i=1}^n v_j^i f(x_i)$ (see (4) below) we obtain $\text{cov}(f(x), u_j) = \sum_{j=1}^n v_j^i k(x, x_i)$. The SGPR variance at $x \in \mathcal{X}$ is $\sigma_m^2(x) = k_m(x, x)$. The computational complexity of obtaining $\nu_m$ and $k_m$ is $O(nm^2)$, which is much smaller than the $O(n^3)$ needed to compute the full posterior for $m \ll n$.

## 2.2 Eigenvector inducing variables

The approximation quality of the SGPR depends on the inducing variables being well chosen. We consider here arguably the conceptually simplest and most canonical sparse approximation based on the eigendecomposition of $\mathbf{K}_{nn}$. Let $\mu_1 \geq \mu_2 \geq \dots \geq \mu_n \geq 0$ be the eigenvalues of the positive semi-definite matrix $\mathbf{K}_{nn}$ with corresponding orthonormal eigenvectors $\mathbf{v}_1, \dots, \mathbf{v}_n$. Writing $\mathbf{v}_j = (v_j^1, \dots, v_j^n)^T \in \mathbb{R}^n$, we take as inducing variables

$$u_j = \mathbf{v}_j^T\mathbf{f} = \sum_{i=1}^n v_j^i f(x_i), \quad j = 1, \dots, m, \tag{4}$$

so that $u_j$ is a linear combination of the inducing points placed at each data point with weights proportional to the entries of the $j^{th}$-largest eigenvector of $\mathbf{K}_{nn}$. Setting $\mathbf{V}_m := [\mathbf{v}_1 \dots \mathbf{v}_m] \in \mathbb{R}^{n \times m}$, this choice of inducing points implies

$$\mathbf{K}_{xm} = \mathbf{k}_n(x)^T\mathbf{V}_m, \qquad \mathbf{K}_{mm} = \text{diag}(\mu_1, \dots, \mu_m), \qquad \mathbf{K}_{nm} = \mathbf{V}_m\text{diag}(\mu_1, \dots, \mu_m) = \mathbf{K}_{mn}^T,$$

see Section C.1 of [5]. Writing $\mathbf{K}_{nn} = \sum_{i=1}^n \mu_i\mathbf{v}_i\mathbf{v}_i^T$ and substituting these expressions into (3), we obtain the SGPR with mean and covariance functions

$$\nu_m(x) = \mathbf{k}_n(x)^T\left[\sum_{k=1}^m \eta_k\mathbf{v}_k\mathbf{v}_k^T\right]\mathbf{y} \tag{5}$$

$$k_m(x, x') = k(x, x') - \mathbf{k}_n(x)^T\left[\sum_{k=1}^m \eta_k\mathbf{v}_k\mathbf{v}_k^T\right]\mathbf{k}_n(x'), \tag{6}$$

where $\eta_k = \frac{1}{\sigma^2+\mu_k}$. We study inferential properties of the SGPR with mean (5) and covariance (6).

Comparing these expressions with those for the full posterior (2), we see that this SGPR is the optimal rank-$m$ approximation in the sense that we have replaced $(\mathbf{K}_{nn} + \sigma^2\mathbf{I}_n)^{-1} = \sum_{i=1}^n \eta_i\mathbf{v}_i\mathbf{v}_i^T$ in both the mean and covariance (2) with $\sum_{i=1}^m \eta_i\mathbf{v}_i\mathbf{v}_i^T$, corresponding to the $m$ largest eigenvalues of $\mathbf{K}_{nn}$. These inducing variables are an example of interdomain inducing features [19], which can yield sparse representations in the spectral domain and computational benefits [14]. Computing this SGPR

involves numerically computing the first $m$ eigenvalues and corresponding eigenvectors of $\mathbf{K}_{nn}$, which can be done in $O(n^2m)$ time using for instance Lanczos iteration [18].

This choice of SGPR using eigenvector inducing variables has been shown to minimize certain upper bounds for the Kullback-Leibler (KL) divergence between order $m$ SGPRs and the true posterior [5], reflecting that it can provide a good posterior approximation. Indeed, one can show that for $m = m_n$ growing fast enough (though sublinearly in $n$), this SGPR converges to the posterior in the KL sense as $n \to \infty$ [5]. Under weaker growth conditions on $m$, this SGPR still converges to the true generative $f_0$ at the minimax optimal rate in the frequentist model [22], even when the SGPR does not converge to the true posterior as $n \to \infty$. Our results include both scenarios and will show that one can still obtain good frequentist UQ performance even for $m$ small enough that the SGPR diverges from the full posterior (see Remark 3.7) unlike in [5].

# 3  Main results

## 3.1  General results on pointwise inference

We now present our main results concerning estimation and uncertainty quantification at a point $x \in \mathcal{X}$ using the SGPR $Q^*$ with eigenvector inducing variables. Since both the posterior and SGPR are GPs, their marginal distributions at $x$ satisfy $f(x)|\mathcal{D}_n \sim \mathcal{N}(\nu_n(x), \sigma_n^2(x))$ and $f(x) \sim \mathcal{N}(\nu_m(x), \sigma_m(x)^2)$, respectively. It thus suffices to study these quantities under the frequentist assumption that there is a 'true' $f_0$ generating the data, i.e. Assumption 2.1. We further define the frequentist bias and variance of the (sparse) posterior mean:

$$b_n(x) = E_{f_0}(\nu_n(x) - f_0(x)), \qquad b_m(x) = E_{f_0}(\nu_m(x) - f_0(x)),$$
$$t_n^2(x) = \text{var}_{f_0}(\nu_n(x)), \qquad t_m^2(x) = \text{var}_{f_0}(\nu_m(x)),$$

where the expectation and variance are taken over $\mathbf{y} \sim P_{f_0}$. We have the following useful relationships between these quantities for the variational posterior and full posterior, which assumes fixed design, so that all statements are taken conditional on the design points $x_1, \ldots, x_n \in \mathcal{X}$. For simplicity, we take the noise variance to be $\sigma^2 = 1$ in (1) for our theoretical results.

**Lemma 3.1.** *For $f_0 \in L^2(\mathcal{X})$, $x \in \mathcal{X}$ and $\mathbf{r}_m(x) := \left[\sum_{j=m+1}^n \eta_j \mathbf{v}_j \mathbf{v}_j^T\right] \mathbf{k}_n(x)$, we have:*

$$b_m(x) = b_n(x) - \langle \mathbf{r}_m(x), \mathbf{f_0} \rangle \tag{7}$$
$$t_m^2(x) = t_n^2(x) - \|\mathbf{r}_m(x)\|^2 \leq t_n^2(x) \tag{8}$$
$$\sigma_m^2(x) = \sigma_n^2(x) + \langle \mathbf{r}_m(x), \mathbf{k}_n(x) \rangle \geq \sigma_n^2(x). \tag{9}$$

The quantity $\mathbf{r}_m(x)$ measures the rank gap due to ignoring the smallest $n - m$ eigenvalues of $\mathbf{K}_{nn}$. The SGPR approximation increases the posterior variance $\sigma_m^2(x) \geq \sigma_n^2(x)$ while reducing the frequentist variance of the posterior mean $t_m^2(x) \leq t_n^2(x)$. Thus if one can control the biases in (7), credible sets from the SGPR will have coverage at least as large as the full posterior. One can therefore ensure good coverage by taking $m = m_n$ sufficiently small, but this can still lead to poor UQ by making the resulting credible intervals extremely wide, and therefore uninformative. We thus first study the effect of $m$ on the quality of estimation, via the posterior convergence rate.

**Proposition 3.2** (Pointwise contraction). *For $f_0 \in C(\mathcal{X})$, $x \in \mathcal{X}$ and $m = m_n \to \infty$,*

$$E_{f_0}Q^*(f : |f(x) - f_0(x)| > M_n \varepsilon_m | \mathcal{D}_n) \to 0,$$

*as $n \to \infty$, where $M_n \to \infty$ is any (arbitrarily slowly growing) sequence and*

$$\varepsilon_m^2 = b_n^2(x) + t_n^2(x) + \sigma_n^2(x) + |\langle \mathbf{r}_m(x), \mathbf{f_0} \rangle|^2 + |\langle \mathbf{r}_m(x), \mathbf{k}_n(x) \rangle|^2.$$

*If $\langle \mathbf{r}_m(x), \mathbf{f_0} \rangle = o(b_n(x))$ and $\langle \mathbf{r}_m(x), \mathbf{k}_n(x) \rangle = o(\sigma_n^2(x))$ as $n \to \infty$, then the rate matches that of the true posterior $\varepsilon_n^2 = b_n^2(x) + t_n^2(x) + \sigma_n^2(x)$.*

The last result says that the SGPR *contracts* about the true $f_0$ at rate $\varepsilon_m$, i.e. it puts all but a vanishingly small amount of probability on functions for which $|f(x) - f_0(x)| \leq M_n \varepsilon_m$, where $f_0$ is the true function generating the data. Such results not only quantify the typical distance between a point estimator $\hat{f}(x)$ (e.g. SGPR mean/median) and the truth ([11], Theorem 8.7), but also the typical

spread of $Q^*$ about the truth. Ideally, most of the $Q^*$-probability should be concentrated on such sets with radius $\varepsilon_m$ proportional to the optimal (minimax) rate, whereupon they have smallest possible size from an information-theoretic perspective.

Contraction rates in global $L^2$-losses have been considered in the variational Bayes literature [24, 2, 41, 27, 26, 22]. Such results crucially rely on the true posterior concentrating exponentially fast about the truth (see e.g. Theorem 5 in [26]), which happens in losses which geometrize the statistical model [15]. This is known *not* to be true for pointwise or uniform loss [15], and hence these previous proof approaches cannot be applied to the present pointwise setting. We must thus use an alternative approach, exploiting the explicit Gaussian structure of the SGPR.

The SGPR approach in UQ is to consider a smallest *pointwise credible set* of probability $1 - \delta$:

$$C_m^\delta = C_m^\delta(\mathcal{D}_n) = [\nu_m(x) - z_{1-\delta}\sigma_m(x), \nu_m(x) + z_{1-\delta}\sigma_m(x)], \tag{10}$$

with $P(|N(0,1)| \leq z_{1-\delta}) = 1 - \delta$. We next establish guarantees on the *frequentist coverage* of $C_m^\delta$, i.e. the probability $1 - \delta'$ with $\inf_{f_0 \in \mathcal{F}} P_{f_0}(f_0(x) \in C_m^\delta) \geq 1 - \delta'$ over a target function class $\mathcal{F}$.

**Remark 3.3** (Zero prior mean). *Since $f_0$ is a-priori unknown, such coverage guarantees must hold uniformly over a function class $\mathcal{F}$ to be meaningful. For symmetric function classes, where $f \in \mathcal{F}$ implies $-f \in \mathcal{F}$, taking a good prior mean $\nu_0$ for $f$ will make estimating $-f$ correspondingly more difficult. Thus without loss of generality we may take zero mean $\nu_0 = 0$.*

**Proposition 3.4** (Pointwise UQ). *Suppose that $\langle \mathbf{r}_m(x), \mathbf{f}_0 \rangle = o(b_n(x))$ as $n \to \infty$. Then for $m = m_n \leq n$ and any $\delta \in (0, 1)$, the frequentist coverage of the credible sets satisfies*

$$\liminf_{n \to \infty} P_{f_0}(f_0(x) \in C_m^\delta) \geq \liminf_{n \to \infty} P_{f_0}(f_0(x) \in C_n^\delta).$$

*If in addition $\|\mathbf{r}_m(x)\|^2 = o(t_n^2(x))$ and $\langle \mathbf{r}_m(x), \mathbf{k}_n(x) \rangle = o(\sigma_n^2(x))$, then*

$$\lim_{n \to \infty} P_{f_0}(f_0(x) \in C_m^\delta) = \lim_{n \to \infty} P_{f_0}(f_0(x) \in C_n^\delta).$$

The first conclusion says that if the rank gap is smaller than the posterior bias (a 'small-bias' condition), SGPR credible sets will have asymptotic coverage at least at the level of the original full posterior, though perhaps larger, including possibly tending to one. The second conclusion gives further conditions on the rank gap under which the asymptotic coverages will be the same. Together with Proposition 3.2, the goal is to obtain a credible set $C_m^\delta$ of smallest possible diameter subject to having sufficient coverage.

## 3.2 Fixed design with rescaled Brownian motion prior

We now apply our general results to a specific GP and find conditions on the number of inducing variables $m$ needed to get good UQ. Consider the domain $\mathcal{X} = [0, 1]$ with regularly spaced design points $x_i = \frac{i}{n+1/2}$ for $i = 1, \ldots, n$. For $B$ a standard Brownian motion, consider as prior $f = \sqrt{c_n}B$, where $c_n = (n + 1/2)^{\frac{1-2\gamma}{1+2\gamma}}$ for $\gamma > 0$. Thus $f$ is a mean-zero GP with covariance kernel $k(x, x') = c_n(x \wedge x')$, where $x \wedge x' := \min(x, x')$. The scaling factor $c_n$ controls the smoothness of the sample paths of the GP and plays the same role as the lengthscale parameter for stationary GPs. The present rescaled Brownian motion is a suitable prior to model Hölder functions of smoothness $\gamma \in (0, 1]$ (e.g. [33, 17, 29]). In particular, for a true $f_0 \in C^\alpha$, $\alpha \in (0, 1]$, one obtains (full) posterior contraction rate $n^{-\frac{\alpha \wedge \gamma}{2\gamma+1}}$ in the global $L^2$-loss [33]. We sometimes write $Q_\gamma^* = Q^*$ for the corresponding SGPR to make explicit that the underlying prior is $\gamma$-smooth.

**Theorem 3.5.** *Let $f_0 \in C^\alpha[0, 1]$, $\alpha \in (0, 1]$ and $x \in (0, 1)$. Consider the SGPR $Q_\gamma^*$ with rescaled Brownian motion prior of regularity $\gamma \in (0, 1]$ and $m = m_n \to \infty$ inducing variables. Then*

$$E_{f_0}Q_\gamma^*(f : |f(x) - f_0(x)| > M_n(n^{-\frac{\alpha \wedge \gamma}{1+2\gamma}} + n^{\frac{2}{1+2\gamma}}m^{-1+\alpha} + n^{\frac{1/2-\gamma}{1+2\gamma}}m^{-3/2})|\mathcal{D}_n) \to 0$$

*as $n \to \infty$, where $M_n \to \infty$ is any (arbitrarily slowly growing) sequence. If $m_n \gg n^{\frac{1}{1+2\gamma}(\frac{2+\alpha}{1+\alpha})}$ then the contraction rate of the SGPR matches that of the full posterior $n^{-\frac{\alpha \wedge \gamma}{1+2\gamma}}$.*

Theorem 3.5 shows that for $m \gg n^{\frac{1}{1+2\gamma}(\frac{2+\alpha}{1+\alpha})}$ inducing points, the SGPR attains the same pointwise contraction rate as the full posterior. If $\gamma = \alpha$, we then recover the optimal (minimax) rate of

convergence for a $C^\alpha$-smooth function. For instance if $\alpha = \gamma = 1$, our result indicates that one can do minimax optimal pointwise estimation based on $m_n \gg \sqrt{n}$ inducing variables, a substantial reduction over the full $n$ observations. Note this convergence guarantee can still hold in cases when the SGPR diverges from the posterior as $n \to \infty$, see Remark 3.7.

The restriction $\alpha \in (0, 1]$ comes from rescaling the underlying Brownian motion [33]. One can extend this to $\alpha > 1$ by considering a smoother baseline process [33], such as integrated Brownian motion, but the more complex form of the resulting eigenvectors and eigenvalues of $\mathbf{K}_{nn}$ make our explicit proof approach difficult. Note also that the prior fixes the value $f(0) = 0$. One can avoid this by adding an independent normal random variable to $f$, but since we consider pointwise inference at a point $x > 0$, this will not affect our results and we thus keep $f(0) = 0$ for simplicity. Nonetheless, the message here is clear: for sufficiently many inducing variables (but polynomially less than $n$), an SGPR based on a $\gamma$-smooth prior can estimate an $\alpha$-smooth truth at rate $n^{-\frac{\alpha \wedge \gamma}{1+2\gamma}}$. We next turn to UQ.

**Theorem 3.6.** *Let $f_0 \in C^\alpha[0, 1]$, $\alpha \in (0, 1]$ and $x \in (0, 1)$. Consider the SGPR $Q_\gamma^*$ with rescaled Brownian motion prior of regularity $\gamma \in (0, 1]$ and $m = m_n \gg n^{\frac{1}{1+2\gamma}(\frac{2+\alpha}{1+\alpha})}$ inducing variables. For $\delta \in (0, 1)$, let $q_m^\delta := P_{f_0}(f_0 \in C_m^\delta(x))$ denote the frequentist coverage of the $1 - \delta$ credible set $C_m^\delta(x)$ given by (10). Then as $n \to \infty$:*

    *(i) (Undersmoothing case) If $\alpha > \gamma$, then $q_m^\delta \to P(|N(0, 1/2)| \le z_{1-\delta}) =: p_\delta > 1 - \delta$ for all $f \in C^\alpha[0, 1]$, where $P(|N(0, 1)| \le z_{1-\delta}) = 1 - \delta$ and $N(0, 1/2)$ is the normal distribution with mean zero and variance $1/2$.*

    *(ii) (Correct smoothing case) If $\alpha = \gamma$, then for each $p \in (0, p_\delta]$, there exists $f \in C^\alpha[0, 1]$ such that $q_m^\delta \to p$.*

    *(iii) (Oversmoothing case) If $\alpha < \gamma$, there exists $f \in C^\alpha[0, 1]$ such that $q_m^\delta \to 0$.*

Theorem 3.6 provides exact expressions for the asymptotic coverage when the credible interval $C_m^\delta$ has diameter $O(n^{-\frac{\alpha \wedge \gamma}{1+2\gamma}})$. In the undersmoothing case $\alpha > \gamma$, the $1 - \delta$ SGPR credible sets are *conservative* from a frequentist perspective, i.e. their coverage converges to a value strictly between $1 - \delta$ and 1. For instance if $\delta = 0.05 \, (0.1)$, the 95% (90%) credible set will have asymptotic coverage 99.4% (98.0%), indicating one does not want to enlarge such credible sets if using enough inducing variables. A desired asymptotic coverage can also be achieved by targetting the credibility according to the formula in Theorem 3.6(i). Reducing $\gamma$ towards zero will generally ensure coverage for Hölder functions, but this will also increase the size of the credible intervals to size $O(n^{-\frac{\gamma}{1+2\gamma}})$, making them less informative. Note that the convergence in (i) is uniform over $C^\alpha$-balls of fixed radius.

In the oversmoothing case $\alpha < \gamma$, the posterior variance underestimates the actual uncertainty and so the credible interval is too narrow, giving overconfident (bad) UQ for many Hölder functions. In the correct smoothing case $\alpha = \gamma$, where we obtain the minimax optimal contraction rate $n^{-\frac{\alpha}{1+2\alpha}}$, coverage falls between these regimes - it does not fully tend to zero, but one can find a function whose coverage is arbitrarily bad, i.e. close to zero.

The best scenario thus occurs when the prior slightly undersmooths the truth, in which case the SGPR credible interval will have slightly conservative coverage but its width $O(n^{-\frac{\gamma}{1+2\gamma}})$ is not too much larger than the minimax optimal size $O(n^{-\frac{\alpha}{1+2\alpha}})$. Our results match the corresponding ones for the full computationally expensive posterior and the main messages are the same: undersmoothing leads to conservative coverage while oversmoothing leads to poor coverage [29].

**Remark 3.7** (Kullback-Leibler). *If the number of inducing variables grows like $n^{\frac{1}{1+2\gamma}(\frac{2+\alpha}{1+\alpha})} \ll m \ll n^{\frac{2}{1+2\gamma}}$, then the conditions of Theorems 3.5 and 3.6 are met, but the KL-divergence between the SGPR and the true posterior tends to infinity as $n \to \infty$ (Lemma A.6 in the supplement). Thus one does not need the SGPR to be an asymptotically exact approximation of the posterior to get similarly good frequentist pointwise estimation and UQ. In particular, one can take $m$ polynomially smaller in $n$ than is required for the KL-divergence to vanish, see [5] for related bounds and discussion.*

While [5] consider approximation quality as measured by the KL-divergence between the SGPR and true posterior, we consider here the different question of whether the SGPR behaves well for pointwise inference, even when it is not a close KL-approximation. The data-generating setup in [5] is also not directly comparable to ours. They take expectations over both the data $(x, y)$ and prior

on $f$, so their results are 'averaged' over the prior, which downweights the effect of sets of small prior probability. In contrast, the frequentist guarantees provided here hold assuming there is a true function $f_0$ which generates the data according to (1) (Assumption 2.1), meaning that one seeks to understand the performance of the method for any fixed $f_0$, including worst-case scenarios. Such setups can lead to very different conclusions, see e.g. Chapters 6.2-6.3 in [11].

## 4 Connections with other Gaussian process priors

The rescaled Brownian motion (rBM) prior we consider here can be thought of as an approximation to the Matérn process of regularity $\gamma \in (0, 1]$ since they both model $\gamma$-smooth functions. The rescaling factor $c_n = (n + 1/2)^{\frac{1-2\gamma}{1+2\gamma}}$ in the covariance function $k(x, x') = c_n(x \wedge x')$ plays the same role as the lengthscale parameter for the Matérn and calibrates the GP smoothness. We thus expect similar theoretical results and conclusions to also hold for the Matérn process, which seems to be the case in practice, as we verify numerically in Section 5. Indeed, similar UQ properties to the rBM posterior are conjectured to hold for the full Matérn posterior [39], in particular that the coverage of credible sets satisfies similar qualitative conclusions to the three cases in Theorem 3.6.

On a more technical level, the success of GPs in nonparametric estimation is known to depend on their sample smoothness, as measured through their small ball probability [33, 36]. The posteriors based on both GPs converge to a $C^\alpha$-truth at rate $n^{-\frac{\alpha \wedge \gamma}{2\gamma+1}}$ in $L^2$-loss ([33] for rBM, [34] for Matérn), indicating the closeness of their small-ball asymptotics and hence that both GPs distribute prior mass similarly over their supports. This gives a more quantitative notion of the similarities of these GPs.

Theoretical guarantees do exist for the Matérn SGPR. Theorem 3.5 of [32] nicely establishes that in a bandit-like setting and if the true $f_0$ is in the prior reproducing kernel Hilbert space $\mathcal{H}$ (roughly $\alpha = \gamma + 1/2$), then inflating credible sets by a factor depending on the rank gap and an upper bound for $\|f_0\|_{\mathcal{H}}$ guarantees coverage. Our parallel results suggest that this inflation is not necessary, since SGPR credible sets in this case will already be conservative. However, since our proofs rely on explicit expressions available for rBM but not for the Matérn, it seems difficult to extend our approach to the different setting considered in [32] (and vice-versa); other techniques are required. Our results can thus be viewed as a guide as to what to expect when using a Matérn SGPR and provide a different perspective reinforcing the main messages of [32], namely that SGPRs can provide reliable, if conservative, UQ.

On the other hand, the squared exponential kernel seems to behave qualitatively differently in numerical simulations, giving different coverage behaviour in the cases considered in Theorem 3.6, see Section 5. This is due to the difference in the smoothness of the sample paths, with squared exponential prior draws being far smoother, in particular analytic. This leads to somewhat different UQ behaviour for the true posterior [12]. Rescaled Brownian motion is a less good approximation for the squared exponential kernel than for the Matérn, and one must thus be cautious about transferring the messages derived here to the squared exponential.

## 5 Numerical simulations

We next investigate empirically the applicability of our theoretical results in finite sample sizes and whether the conclusions extend to related settings, such as different designs and other GP priors. We consider various nonparametric regression settings of the form (1) with $C^\alpha$-smooth truths and $\gamma$-smooth priors. We compute both the full posterior (columns marked GP) and SGPR (marked SGPR) with $m \ll n$ eigenvector inducing variables for $f(x_0)$ at a fixed point $x_0 \in \mathcal{X}$, and report the root-mean square error (RMSE), negative log predictive density (NLPD), and the length and coverage of the 90%-credible intervals of the form (10) (i.e. $\delta = 0.1$), see Section B in the supplement for definitions. The true noise variance in (1) is taken to be $\sigma^2 = 1$, but is considered unknown and is estimated by maximizing the log-marginal likelihood as usual. We ran all simulations 500 times and report average values and their standard deviations when relevant. Simulations were run on a 2 GHz Quad-Core Intel Core i5 processor on a 2020 Macbook Pro with 16GB of RAM.

**Settings of the theoretical results.** We consider the setting of our theoretical results in Section 3 with $\mathcal{X} = [0, 1]$ and $x_i = i/(n + 1/2)$ for $i = 1, \ldots, n$. To generate the data, we take $f_0(x) = |x - 0.5|^\alpha$, which is exactly $\alpha$-Hölder at $x_0 = 0.5$, and investigate pointwise inference at $x_0 = 0.5$ for

different choices of $\alpha$. As Gaussian priors, we consider (i) rescaled Brownian motion of regularity $\gamma$ described in Section 3, (ii) the Matérn kernel with parameter $\gamma$ and (iii) the square exponential kernel $k(x,x') = \exp\{-\frac{1}{2\ell_n^2}(x-x')^2\}$, where the lengthscale $\ell_n = n^{-\frac{1}{1+2\gamma}}$ is chosen to be appropriate for estimating $\gamma-$Hölder smooth functions [33]. We take $m^* := n^{\frac{1}{1+2\gamma}\frac{2+\alpha}{1+\alpha}}$ inducing variables for the SGPR. Results are given in Table 1: **Fixed Design, $n = 1000$**. We also consider the related case of uniform random design where $x_i \sim U(0,1)$ (Table 1: **Random Design, $n = 500$**).

| **Prior** | **Coverage** | | **Length** | | **RMSE** | | **NLPD** | |
|---|---|---|---|---|---|---|---|---|
| GP | SGPR | GP | SGPR | GP | SGPR | GP | SGPR | GP |
| **Fixed Design:** $n = 1000$, $(\alpha, \gamma) = (1.0, 0.5)$ | | | | | | | | |
| rBM | 0.98 | 0.98 | 0.41 | 0.41 | 0.09 | 0.09 | -0.90 (0.21) | -0.90 (0.21) |
| Matérn | 0.98 | 0.98 | 0.49 | 0.49 | 0.10 | 0.10 | -0.68 (0.16) | -0.68 (0.16) |
| SE | 0.91 | 0.91 | 0.65 | 0.65 | 0.19 | 0.19 | -0.28 (0.30) | -0.28 (0.30) |
| **Fixed Design:** $n = 1000$, $(\alpha, \gamma) = (0.5, 0.5)$ | | | | | | | | |
| rBM | 0.74 | 0.74 | 0.41 | 0.41 | 0.18 | 0.18 | -0.09 (0.35) | -0.09 (0.35) |
| Matérn | 0.84 | 0.84 | 0.49 | 0.49 | 0.17 | 0.17 | -0.40 (0.32) | -0.40 (0.32) |
| SE | 0.88 | 0.88 | 0.65 | 0.65 | 0.21 | 0.21 | -0.21 (0.31) | -0.21 (0.31) |
| **Random Design:** $n = 500$, $(\alpha, \gamma) = (1.0, 0.5)$ | | | | | | | | |
| rBM | 0.98 | 0.98 | 0.49 (0.02) | 0.49 (0.02) | 0.11 | 0.11 | -0.65 (0.15) | -0.65 (0.15) |
| Matérn | 0.96 | 0.96 | 0.59 (0.03) | 0.59 (0.03) | 0.13 | 0.13 | -0.55 (0.14) | -0.55 (0.14) |
| SE | 0.92 | 0.92 | 0.76 (0.08) | 0.76 (0.08) | 0.20 | 0.20 | -0.02 (0.25) | -0.02 (0.25) |
| **Random Design:** $n = 500$, $(\alpha, \gamma) = (0.3, 0.5)$ | | | | | | | | |
| rBM | 0.25 | 0.25 | 0.49 (0.02) | 0.49 (0.02) | 0.37 | 0.37 | 2.23 (0.86) | 2.23 (0.86) |
| Matérn | 0.47 | 0.47 | 0.59 (0.02) | 0.59 (0.02) | 0.34 | 0.34 | 0.91 (0.66) | 0.91 (0.66) |
| SE | 0.71 | 0.71 | 0.77 (0.08) | 0.77 (0.08) | 0.31 | 0.31 | 0.36 (0.52) | 0.36 (0.52) |

Table 1: Comparison of SGPR and full posterior (marked GP) for 90% pointwise credible intervals for different values of $(\alpha, \gamma)$. For the SGPR we use $m^* := n^{\frac{1}{1+2\gamma}\frac{2+\alpha}{1+\alpha}}$ inducing variables, corresponding to 178, 316, 106 and 244 in the four $(n, \alpha, \gamma)$ regimes above (top-to-bottom).

Theorem 3.6 predicts that for 90% credible sets ($\delta = 0.1$), $m \gg m^*$ and $\alpha > \gamma$, the coverage should converge to $p_{0.1} = P(|N(0, 1/2)| < z_{0.9}) = 0.98$. We see that for $n = 1000$, the observed coverages for both the full posterior and SGPR based on the Brownian motion prior are very close to this predicted value, and are indeed conservative. In the oversmoothing case $\alpha < \gamma$, coverage is poor and far below the nominal level, while in the correct smoothing case ($\alpha = \gamma$), coverage is moderately below the nominal level. We see that the asymptotic theory is applicable for reasonable samples sizes.

The Matérn process behaves qualitatively similarly to Brownian motion in these three cases, though the (still conservative) limiting coverage in the undersmoothing case is predicted to be slightly different [39], as reflected in Table 1. On the other hand, the square exponential behaves differently in all cases and generally has wider sets with this lengthscale choice. The wider credible intervals and larger RMSE suggest greater bias of the posterior mean. While rescaled Brownian motion seems a reasonable guide for the Matérn, it does not seem so for the squared exponential, see Section 4.

For all three GP priors, the reported metrics are practically indistinguishable between the SGPR and the full GP, confirming that $m \gg n^{\frac{1}{1+2\gamma}\frac{2+\alpha}{1+\alpha}}$ inducing variables seems sufficient to obtain virtually the same pointwise credible intervals as the full posterior. However, since $m^* \ll n^{\frac{2}{1+2\gamma}}$ for any $\alpha > 0$, the KL-divergence between the SGPR and the full posterior grows to infinity as $n \to \infty$ in the Brownian motion case (Lemma A.6), and hence the SGPR will differ from the posteriors in some ways. This reflects that one does not necessarily need to fully approximate the posterior in order to match its pointwise inference properties. Note that for fixed design, the length of the credible sets is not random since it depends only on the features $x_1, \ldots, x_n$, and so no standard errors are reported.

**Other settings.** We turn to simulation settings not covered by our theory. Given the above discussion, we consider only the Matérn process in dimension $d = 10$ and the corresponding SGPR with $m = m_d^* := n^{\frac{d}{d+2\gamma}}$. As features, consider random design with (i) $x_i \sim \mathcal{U}([-1/2, 1/2]^d)$ and (ii) $X_i \sim \mathcal{N}_d(0, \Sigma_\rho)$ for $[\Sigma_\rho]_{ij} = 1$ for $i = j$ and $[\Sigma_\rho]_{ij} = \rho$ otherwise. To better reflect correlation in real-world data, we also consider a semi-synthetic dataset with real features but simulated responses.

As features, we took the first $d = 10$ columns and randomly sampled $n = 2000$ rows from a Korean meteorological dataset [8]. In all cases, we consider estimating the $\alpha-$Hölder function $f_0(x) = \|x - x_0\|^{\alpha}$ at $x_0 = 0$, and the resulting pointwise 90%-credible sets at $x_0$ coming from the Matérn SGPR. Results are presented in Table 2, where we see that for $m = m_d^*$ the performance of the pointwise credible sets from the SGPR matches that of the full posterior in each design. Due to the higher dimension, credible intervals are less confident (wider) and have larger coverage.

**Multidimensional Random Design, $n = 1000$**

| Design | | Smoothness | | Coverage | | Length | | NLPD | |
|--------|------|------|------|------|------|------|------|------|------|
| Type | $\rho$ | $\alpha$ | $\gamma$ | SGPR | GP | SGPR | GP | SGPR | GP |
| Uniform | | 0.5 | 0.5 | 0.96 | 0.96 | 1.46 (0.03) | 1.46 (0.03) | 0.96 (0.26) | 0.96 (0.26) |
| Gaussian | 0.0 | 0.7 | 0.5 | 0.99 | 0.99 | 2.17 (0.05) | 2.17 (0.05) | 1.34 (0.11) | 1.34 (0.11) |
| Gaussian | 0.2 | 0.9 | 0.5 | 1.00 | 1.00 | 2.13 (0.06) | 2.13 (0.06) | 1.90 (0.14) | 1.90 (0.14) |
| Gaussian | 0.5 | 1.1 | 0.5 | 1.00 | 1.00 | 2.03 (0.06) | 2.03 (0.06) | 1.36 (0.12) | 1.36 (0.12) |

**Semi-synthetic Data, $n = 2000$**

| Smoothness | | Coverage | | Length | | RMSE | | NLPD | |
|------|------|------|------|------|------|------|------|------|------|
| $\alpha$ | $\gamma$ | SGPR | GP | SGPR | GP | SGPR | GP | SGPR | GP |
| 1.0 | 0.5 | 1.00 | 1.00 | 2.29 | 2.29 | 0.16 | 0.16 | 3.22 (0.47) | 3.22 (0.47) |

Table 2: Comparison of SGPR and full posterior (GP) with Matérn prior for 90% pointwise credible intervals for different values of $(\alpha, \gamma)$. The SGPR uses 534 and 1002 inducing variables in the first four rows and fifth row, respectively.

## 6 Discussion

In this work, we established the frequentist asymptotic behaviour of pointwise credible intervals coming from sparse variational Gaussian process regression (SGPR) with eigenvector inducing variables based on a rescaled Brownian motion prior. We showed that if the prior undersmooths the truth and with enough inducing variables, SGPR credible sets can provide reliable, though conservative, uncertainty quantification, with the coverage converging to a value strictly between the nominal level and one. If the prior oversmooths the truth, UQ can be poor. We further showed that it is not necessary for the SGPR to converge to the true posterior in KL-divergence to have similar behaviour of the credible sets. Our results suggest that properly calibrated SGPRs can perform reliable UQ. We verified these conclusions in simulations and discussed connections with other GPs.

Despite the widespread use of SGPR, there are still relatively few theoretical guarantees for their use, particularly for UQ. Our work provides some new results in this direction, but most of this research area is still wide open. A key step would be to prove similar results for the most commonly used GP priors, notably the Matérn and squared exponential. Similarly, one would like to extend these results to other choices of inducing variables, for instance the eigenfunctions of the kernel operator. While our results give some intuition of what one can expect, they rely on a careful analysis of Brownian motion with fixed design, and new ideas and techniques will be needed for these other settings.

It is also unclear what minimal number of inducing variables is needed to get good pointwise UQ. When $\alpha = \gamma$, the threshold for minimax convergence rates for *estimation* in $L^2$ is $n^{\frac{1}{1+2\alpha}}$ [22], which is smaller than our bound $n^{\frac{1}{1+2\alpha}(\frac{2+\alpha}{1+\alpha})}$, see Section B in the supplement for some related numerical simulations. Our results are also *non-adaptive* since they assume fixed prior smoothness $\gamma$, whereas one often selects $\gamma$ in a data-driven way, for instance by maximizing the marginal likelihood or evidence lower bound (ELBO). However, *adaptive* UQ is a subtle topic which typically requires further assumptions (e.g. self-similarity) on the true generative function to even be possible [30], and extending our results to such settings will require significant technical and conceptual work.

**Acknowledgements.** We are grateful to four anonymous reviewers for helpful comments that improved the manuscript. The authors would like to thank the Imperial College London-CNRS PhD Joint Programme for funding which supported Luke Travis in his studies.

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

# Supplementary material to "Pointwise uncertainty quantification for sparse variational Gaussian process regression with a Brownian motion prior"

## A  Proofs

Throughout this section, for sequences $a_n$ and $b_n$, we will write $a_n \lesssim b_n$ if there exists $C > 0$ such that $a_n \leq C b_n$ for $n$ large enough, and we will write $a_n \asymp b_n$ if both $a_n \lesssim b_n$ and $b_n \lesssim a_n$. Recall that for simplicity, we take the noise variance $\sigma^2 = 1$ in the regression model (1).

### A.1  General results on pointwise inference

*Proof of Lemma 3.1.* Using the definitions (2) and (3) of the posterior and SGPR means, we have $\nu_m(x) = \nu_n(x) - \langle \mathbf{r}_m(x), \mathbf{y} \rangle$, from which (7) follows on taking the $E_{f_0}$-expectation. Next, recall that $\nu_n(x) = \mathbf{k}_n(x)^T \left[ \sum_{j=1}^n \eta_j \mathbf{v}_j \mathbf{v}_j^T \right] \mathbf{y}$, and thus

$$
\begin{aligned}
t_n^2(x) = \mathrm{Var}(\nu_n(x)) &= \left\| \left[ \sum_{j=1}^n \eta_j \mathbf{v}_j \mathbf{v}_j^T \right] \mathbf{k}_n(x) \right\|^2 = \left\| \left[ \sum_{j=1}^m \eta_j \mathbf{v}_j \mathbf{v}_j^T \right] \mathbf{k}_n(x) + \mathbf{r}_m(x) \right\|^2 \\
&= \left\| \left[ \sum_{j=1}^m \eta_j \mathbf{v}_j \mathbf{v}_j^T \right] \mathbf{k}_n(x) \right\|^2 + \| \mathbf{r}_m(x) \|^2 \\
&= t_m^2(x) + \| \mathbf{r}_m(x) \|^2,
\end{aligned}
$$

from which (8) follows. Finally,

$$
\begin{aligned}
\sigma_n^2(x) = k_n(x, x) &= k(x, x) - \mathbf{k}_n(x)^T \left[ \sum_{j=1}^n \eta_j \mathbf{v}_j \mathbf{v}_j^T \right] \mathbf{k}_n(x) \\
&= k(x, x) - \mathbf{k}_n(x)^T \left[ \sum_{j=1}^m \eta_j \mathbf{v}_j \mathbf{v}_j^T \right] \mathbf{k}_n(x) - \mathbf{k}_n(x)^T \left[ \sum_{j=m+1}^n \eta_j \mathbf{v}_j \mathbf{v}_j^T \right] \mathbf{k}_n(x),
\end{aligned}
$$

which gives (9).

$\square$

*Proof of Proposition 3.2.* We will prove this for the full posterior, with the expressions for the SGPR following the same. Denoting by $E_{\Pi|Y}$ the expectation with respect to the posterior, Markov's inequality yields,

$$
\Pi \left( |f(x) - f_0(x)| > M_n \varepsilon_n | Y \right) \leq \frac{E_{\Pi|Y} \left( f(x) - f_0(x) \right)^2}{M_n^2 \varepsilon_n^2}.
$$

Moreover,

$$
E_{\Pi|Y} \left( f(x) - f_0(x) \right)^2 = \left( \nu_n(x) - f_0(x) \right)^2 + \mathrm{Var}_{\Pi|Y} \left( f(x) \right) = \left( \nu_n(x) - f_0(x) \right)^2 + \sigma_n^2(x).
$$

Now,

$$
E_{f_0} \left[ E_{\Pi|Y} \left( f(x) - f_0(x) \right)^2 \right] = E_{f_0} \left[ (\nu_n(x) - f_0(x))^2 \right] + \sigma_n^2(x) = b_n^2(x) + t_n^2(x) + \sigma_n^2(x).
$$

Thus for $\varepsilon_n^2(x) \asymp b_n^2(x) + t_n^2(x) + \sigma_n^2(x)$ and any $M_n \to \infty$, $E_{f_0} \Pi \left( |f(x) - f_0(x)| > M_n \varepsilon_n | Y \right) \to 0$.

$\square$

*Proof of Proposition 3.4.* We have,

$$
P_{f_0}(f_0(x) \in C_m^\delta) = P_{f_0}(|\nu_m(x) - f_0(x)| \leq z_{1-\delta} \sigma_m(x)) = P_{f_0}(|V_m| \leq z_{1-\delta}),
$$

for $V_m \sim \mathcal{N}\left(\frac{b_m(x)}{\sigma_m(x)}, \frac{t_m^2(x)}{\sigma_m^2(x)}\right)$. Now, if $\langle \mathbf{r}_m(x), \mathbf{f}_0 \rangle = o(b_n(x))$, we have $b_m(x) = b_n(x)(1 + o(1))$ and further, by expressions (8) and (9) we have $\frac{t_m^2(x)}{\sigma_m^2(x)} \leq \frac{t_n^2(x)}{\sigma_n^2(x)}$, and thus,

$$\liminf_{n \to \infty} P_{f_0}\left(|V_m| \leq z_{1-\delta}\right) \geq \liminf_{n \to \infty} P_{f_0}\left(|\mathbf{V}_n| \leq z_{1-\delta}\right) = \liminf_{n \to \infty} P_{f_0}\left(f_0(x) \in C_n^\delta\right).$$

If in addition, we have $\|\mathbf{r}_m(x)\|^2 = o(t_n^2(x))$ and $\langle \mathbf{r}_m(x), \mathbf{k}_n(x) \rangle = o(\sigma_n^2(x))$ then by expressions (8) and (9) we have $t_m^2(x) = t_n^2(x)(1 + o(1))$ and $\sigma_m^2(x) = \sigma_n^2(x)(1 + o(1))$ and thus,

$$\lim_{n \to \infty} P_{f_0}\left(|V_m| \leq z_{1-\delta}\right) = \lim_{n \to \infty} P_{f_0}\left(|\mathbf{V}_n| \leq z_{1-\delta}\right) = \lim_{n \to \infty} P_{f_0}\left(f_0(x) \in C_n^\delta\right).$$

$\square$

## A.2 Rescaled Brownian motion prior

Recall that we take regularly spaced design points $x_i = \frac{i}{n+1/2}$, $i = 1, \ldots, n$, and that we consider the rescaled Brownian motion prior, which is a mean-zero Gaussian process with covariance kernel $k(x, x') = c_n(x \wedge y) = c_n \min(x, y)$ with $c_n = (n+1/2)^{\frac{1-2\gamma}{1+2\gamma}}$ for $\gamma > 0$. For notational convenience, we write $N = (n+1/2)/c_n = (n+1/2)^{1-\frac{1-2\gamma}{1+2\gamma}}$. The kernel matrix $\mathbf{K}_{nn}$ evaluated at the sample points then equals

$$\mathbf{K}_{nn} = c_n \begin{pmatrix} 1/(n+1/2) & 1/(n+1/2) & \cdots & 1/(n+1/2) \\ 1/(n+1/2) & 2/(n+1/2) & \cdots & 2/(n+1/2) \\ \vdots & \vdots & \ddots & \vdots \\ 1/(n+1/2) & 2/(n+1/2) & \cdots & n/(n+1/2) \end{pmatrix} = \frac{1}{N} \begin{pmatrix} 1 & 1 & \cdots & 1 \\ 1 & 2 & \cdots & 2 \\ \vdots & \vdots & \ddots & \vdots \\ 1 & 2 & \cdots & n \end{pmatrix}.$$

For this specific choice of prior and design, we can compute fairly explicit expressions for the eigenvalues and eigenvectors of $\mathbf{K}_{nn}$, which will in turn allow us to obtain the precise pointwise asymptotics of the VB method needed for our results.

**Lemma A.1** (Eigenvalues and eigenvectors). *The eigenvalues $(\mu_j)_{j=1}^n$ of the covariance matrix $\mathbf{K}_{nn}$ are given by*

$$\mu_j = \frac{1}{2N(1 - \cos(\psi_j))} \qquad \text{with} \qquad \psi_j = \frac{j - 1/2}{n + 1/2}\pi$$

*for $j = 1, \ldots, n$. Moreover, their size satisfies*

$$\mu_j \asymp \frac{1}{N\psi_j^2} \asymp (n+1/2)c_n \frac{1}{(j-1/2)^2}.$$

*The corresponding orthonormal eigenvectors $\mathbf{v}_1, \ldots, \mathbf{v}_n$, where $\mathbf{v}_j = (v_j^1, \ldots, v_j^n)^T$, are given by*

$$v_j^l = \frac{2\sin(l\psi_j)}{\sqrt{2n+1}}.$$

*Proof.* Let $v$ be an eigenvector of $\mathbf{K}_{nn}$ with corresponding eigenvalue $\mu$, so that $\mathbf{K}_{nn}v = \mu v$. Applying elementary row operations to this equation, we arrive at the following equivalent equations:

$$\frac{1}{N} \begin{bmatrix} v^1 \\ v^2 \\ \vdots \\ v^{n-1} \\ v^n \end{bmatrix} = \mu \begin{bmatrix} 2v^1 - v^2 \\ 2v^2 - v^1 - v^3 \\ \vdots \\ 2v^{n-1} - v^{n-2} - v^n \\ v^n - v^{n-1} \end{bmatrix}.$$

From this, one can see that to obtain a non-trivial eigenvector $v \neq 0$, we require $v^1 \neq 0$. We thus consider the unnormalized eigenvector $w = (w^1, \ldots, w^n)^T \in \mathbb{R}^n$, for which we set $w^1 = 1$ for simplicity. Further define $w^0 = 0$ and $w^{-1} = -1$, so that the first $(n-1)$ equations in the last display are equivalent (after rearranging) to the recurrence relation

$$w^j = \left(2 - \frac{1}{\mu N}\right) w^{j-1} - w^{j-2}, \qquad j = 1, \ldots, n, \tag{11}$$

while the last equation gives the boundary condition

$$w^n = \frac{\mu N}{\mu N - 1} w^{n-1}.$$ (12)

The linear recurrence relation (11) has solution

$$w^j = \frac{1}{\phi_+ - \phi_-} \left( \phi_+^j - \phi_-^j \right),$$ (13)

for

$$\phi_\pm = 1 - \frac{1}{2\mu N} \pm \frac{1}{2} \sqrt{\frac{1}{\mu^2 N^2} - \frac{4}{\mu N}} =: x \pm iy,$$

with $x = 1 - \frac{1}{2\mu N}$ and $y = \frac{1}{2}\sqrt{\frac{4}{\mu N} - \frac{1}{\mu^2 N^2}}$ (computation of the eigenvalues below reveals that $\frac{1}{\mu^2 N^2} - \frac{4}{\mu N} < 0$ for all eigenvalues $\mu$.) Note that (13) provides the solution to (11) for any arbitrary value of $\mu$, but that (12) will later be needed to ensure that $\mu$ is an eigenvalue of $\mathbf{K}_{nn}$. Since $x^2 + y^2 = 1$, we can conveniently rewrite this in complex exponential form as

$$\phi_\pm = e^{\pm i\psi}, \qquad \text{where} \qquad \cos\psi = \text{Re}(\phi_\pm) = x = 1 - \frac{1}{2\mu N},$$ (14)

and $\psi$ is an alternative parametrization of $\mu$. This form makes it easy to compute expressions for $\phi_\pm^j$, and hence evaluate (13). Indeed, we have

$$w^j = \frac{1}{\phi_+ - \phi_-}(\phi_+^j - \phi_-^j) = \frac{\sin(j\psi)}{\sin\psi},$$ (15)

giving the form of the unnormalized eigenvector $w = (1, w^2, \ldots, w^n)^T$ corresponding to an eigenvalue $\mu$ via (14). To normalize the eigenvector, using that $\cos(2j\psi) = 1 - 2\sin^2(j\psi)$,

$$\|w\|_2^2 = \sum_{j=1}^n (w^j)^2 = \sum_{j=1}^n \frac{\sin^2(j\psi)}{\sin^2\psi} = \frac{1}{\sin^2\psi} \sum_{j=1}^n \frac{1}{2}(1 - \cos(2j\psi))$$

$$= \frac{1}{4\sin^2\psi}\left(2n + 1 - \frac{\sin((2n+1)\psi)}{\sin\psi}\right),$$

where in the last line we have used the Lagrange trigonometric identity that $\sum_{j=0}^n \cos(2j\psi) = \frac{1}{2} + \frac{\sin(2n+1)\psi)}{2\sin\psi}$ for any $\psi \neq 0 \pmod \pi$ (else it equals $n + 1$). On computing the values $\psi$ corresponding to the eigenvalues in (17) below, one has $\sin((2n+1)\psi) = 2\sin((n+1/2)\psi)\cos((n+1/2)\psi) = 0$, and hence $\|w\|_2^2 = \frac{2n+1}{4\sin^2\psi}$. Thus the normalized eigenvector with $v = w/\|w\|_2$ has the required coordinates,

$$v^j = \frac{2\sin(j\psi)}{\sqrt{2n+1}}.$$ (16)

Turning to the computation of the eigenvalues, the boundary condition (12) defines whether $\mu$ is an eigenvalue. Substituting the solution (15) into (12), any $\mu$ that satisfies

$$\frac{\sin(n\psi)}{\sin\psi} = \frac{\mu N}{\mu N - 1}\frac{\sin((n-1)\psi)}{\sin\psi},$$

or equivalently

$$(\mu N - 1)\sin(n\psi) = \mu N \sin((n-1)\psi) \qquad \text{and} \qquad \sin\psi \neq 0,$$

is an eigenvalue of $\mathbf{K}_{nn}$. Indeed, if one approaches the problem in the traditional way, by computing the determinant of $(\mathbf{K}_{nn} - \mu\mathbf{I}_n)$ (see Lemma A.2 below), then one has

$$|\mathbf{K}_{nn} - \mu\mathbf{I}_n| = \frac{(-1)^n}{\sin\psi}\left(\left[\mu^n - \frac{\mu^{n-1}}{N}\right]\sin(n\psi) - \mu^n\sin((n-1)\psi)\right) = 0$$

$$\Leftrightarrow \left[\mu - \frac{1}{N}\right]\sin(n\psi) - \mu N \sin((n-1)\psi) = 0 \qquad \text{and} \qquad \sin\psi \neq 0$$

$$\Leftrightarrow (\mu N - 1)\sin(n\psi) - \mu N \sin((n-1)\psi) = 0 \qquad \text{and} \qquad \sin\psi \neq 0,$$

exactly as above. Recalling that $\psi$ and $\mu$ are related by $\cos\psi = 1 - \frac{1}{2\mu N}$, this last expression is equivalent to

$$
\begin{aligned}
0 &= \left(1 - \frac{1}{\mu N}\right)\sin(n\psi) - \sin((n-1)\psi) \\
&= (2\cos\psi - 1)\sin(n\psi) - \sin((n-1)\psi) \\
&= \sin((n+1)\psi) - \sin(n\psi) \\
&= 2\sin(\psi/2)\cos((n+1/2)\psi),
\end{aligned}
$$

where we have used the identity $2\cos A\sin B = \sin(A+B) - \sin(A-B)$ twice. By (14), the eigenvalues $\mu$ are only defined by $\cos\psi$, and hence we may restrict $\psi$ to $[-\pi, \pi)$ and look for solutions within this range. For $\psi$ in this range, the condition $\sin\psi \neq 0$ then implies $\sin(\psi/2) \neq 0$, and hence we need only look for solutions to the equation $\cos((n+1/2)\psi) = 0$. For $\psi \in [-\pi, \pi)$, these are given by

$$
\psi_j = \frac{j - 1/2}{n + 1/2}\pi, \quad j = -n, -n+1, \ldots, n. \tag{17}
$$

By symmetry, $\cos\psi_j = \cos\psi_{1-j}$ for $j = 1, \ldots, n$ and hence we may discard the roots $j = -n+1, \ldots, 0$. Moreover, $\sin\psi_{-n} = 0$ violating the restriction $\sin\psi \neq 0$. This leaves us with the $n$ eigenvalues

$$
\mu_j = \frac{1}{2N(1 - \cos\psi_j)}, \quad j = 1, \ldots, n
$$

as desired.

It remains only to quantify the size of the eigenvalues $\mu_j$. Expanding $\cos x = 1 - x^2/2! + x^4/4! + o(x^4)$, we have $x^2 - x^4/12 \leq 2(1 - \cos x) \leq x^2$ for all $x$, and thus

$$
\frac{1}{Nx^2} \leq \frac{1}{2N(1 - \cos x)} \leq \frac{1}{Nx^2(1 - x^2/12)}.
$$

Since $\psi_j = \frac{j-1/2}{n+1/2}\pi$, we have $\psi_j^2/12 < 5/6$ (on bounding $\pi^2$ by 10 and the fraction by 1) and so $\frac{1}{N\psi_j^2(1 - \psi_j^2/12)} < \frac{6}{N\psi_j^2}$. Hence we have $\frac{1}{N\psi_j^2} \leq \mu_j \leq \frac{6}{N\psi_j^2}$ and $\mu_j \asymp \frac{1}{N\psi_j^2}$. $\qquad\square$

Let $|A|$ denote the determinant of a matrix $A$.

**Lemma A.2.** *The solutions to the characteristic equation of $\mathbf{K}_{nn}$, $|\mathbf{K}_{nn} - \mu\mathbf{I}_n| = 0$, are given by the solutions to*

$$
\frac{(-1)^n}{\sin\psi}\left(\left(\mu^n - \frac{1}{N}\mu^{n-1}\right)\sin(n\psi) - \mu^n\sin((n-1)\psi)\right) = 0,
$$

*where $\cos\psi = 1 - \frac{1}{2N\mu}$.*

*Proof.* Using row operations, the desired determinant equals

$$
|\mathbf{K}_{nn} - \mu\mathbf{I}_n| = 
\begin{vmatrix}
\frac{1}{N} - \mu & \frac{1}{N} & \cdots & \frac{1}{N} \\
\frac{1}{N} & \frac{2}{N} - \mu & \cdots & \frac{2}{N} \\
\vdots & \vdots & \ddots & \vdots \\
\frac{1}{N} & \frac{2}{N} & \cdots & \frac{n}{N} - \mu
\end{vmatrix}
=
\begin{vmatrix}
\frac{1}{N} - \mu & \frac{1}{N} & \cdots & \cdots & \frac{1}{N} \\
\mu & \frac{1}{N} - \mu & \frac{1}{N} & \cdots & \frac{1}{N} \\
0 & \mu & \ddots & \ddots & \vdots \\
\vdots & & \ddots & \frac{1}{N} - \mu & \frac{1}{N} \\
0 & \cdots & 0 & \mu & \frac{1}{N} - \mu
\end{vmatrix},
$$

which can be further be reduced to

$$
|\mathbf{K}_{nn} - \mu\mathbf{I}_n| = 
\begin{vmatrix}
\frac{2}{N} - \mu & \mu & 0 & \cdots & 0 \\
\mu & \frac{2}{N} - \mu & \mu & \ddots & \vdots \\
0 & \mu & \ddots & \ddots & 0 \\
\vdots & & \ddots & \frac{2}{N} - \mu & \mu \\
0 & \cdots & 0 & \mu & \frac{1}{N} - \mu
\end{vmatrix}
=: d_n,
$$

the determinant of a symmetric tridiagonal $n \times n$ matrix. Using the Laplace expansion of the determinant in terms of the determinants of its $(n-1) \times (n-1)$ submatrices, we obtain the recurrence relation $d_n = \left(\frac{2}{N} - \mu\right) d_{n-1} - \mu^2 d_{n-2}$. The solution to this recurrence is given by

$$d_n = \frac{1}{\phi_+ - \phi_-} \left(\phi_+^{n+1} - \phi_-^{n+1}\right),$$

where $\phi_\pm$ satisfy $\phi^2 - \left(\frac{2}{N} - \mu\right)\phi + \mu^2 = 0$. That is,

$$\phi_\pm = \frac{\left(\frac{2}{N} - \mu\right) \pm \sqrt{\frac{4}{N^2} - \frac{4\mu}{N}}}{2}.$$

Similar to when we solved the recurrence in the proof of Lemma A.1, we end up with $\phi_\pm = \mu e^{\pm i\psi'}$, where $\cos(\psi') = \frac{1}{2N\mu} - 1$. We use the notation $\psi'$ to emphasise that these angles are different to the $\psi$'s found in the proof of Lemma A.1; indeed, it is easy to see that $\psi' = \psi + \pi$, so that $\sin(n\psi') = (-1)^n \sin(n\psi)$. This yields

$$d_n = (-1)^n \frac{\mu^n}{\sin(\psi)} \sin((n+1)\psi),$$

and the result is shown by seeing

$$|\mathbf{K}_{nn} - \mu I| = \left(\frac{1}{N} - \mu\right) d_{n-1} - \mu^2 d_{n-2}$$

$$= (-1)^{n-1}\left(\frac{1}{N} - \mu\right)\frac{\mu^{n-1}}{\sin(\psi)}\sin(n\psi) - (-1)^{n-2}\mu^2 \frac{\mu^{n-2}}{\sin(\psi)}\sin((n-1)\psi)$$

$$= \frac{(-1)^n}{\sin\psi}\left(\left(\mu^n - \frac{1}{N}\mu^{n-1}\right)\sin(n\psi) - \mu^n \sin((n-1)\psi)\right)$$

$$\square$$

Given the explicit expressions derived for the eigenvalues and eigenvectors of the covariance matrix $\mathbf{K}_{nn}$, we derive bounds for the size of the remainder terms presented in Lemma 3.1.

**Lemma A.3.** *Suppose that $f_0 \in C^\alpha[0,1]$, $\alpha \in (0,1]$, and consider the rescaled Brownian motion prior of regularity $\gamma \in (0,1]$. The corresponding SGPR $Q^*$ based on the first $m = m_n \to \infty$ eigenvector inducing variables satisfies, as $n \to \infty$,*

$$b_m(x) = b_n(x) + O(nc_n m^{-(1+\alpha)}) \tag{18}$$

$$t_m^2(x) = t_n^2(x) + O(nc_n^2 m^{-3}) \tag{19}$$

$$\sigma_m^2(x) = \sigma_n^2(x) + O(nc_n^2 m^{-3}). \tag{20}$$

*Proof.* First, suppose that $x = x_i$ is a design point. Starting with the bias, we have,

$$b_m(x) = b_n(x) + \mathbf{k}_n(x)^T \left(\sum_{j=m+1}^n \eta_j \mathbf{v}_j \mathbf{v}_j^T\right) \mathbf{f}_0 = b_n(x) + \sum_{j=m+1}^n \eta_j (\mathbf{k}_n(x)^T \mathbf{v}_j)(\mathbf{v}_j^T \mathbf{f}_0).$$

In this case $\mathbf{k}_n(x_i)^T \mathbf{v}_j = \mu_j v_j^i$ and the next step is to bound $\mathbf{v}_j^T \mathbf{f}_0$ for each $j$. Using the explicit form for $\mathbf{v}_j$ given in Lemma A.1,

$$\mathbf{v}_j^T \mathbf{f}_0 = \sum_{l=1}^n \frac{2}{\sqrt{2n+1}} \sin(l\psi_j) f_0(x_l) = \frac{2}{\sqrt{2n+1}} \sum_{l=1}^n \sin\left(l\frac{(j-1/2)\pi}{n+1/2}\right) f_0(x_l)$$

$$= \frac{2}{\sqrt{2n+1}} \sum_{l=1}^n \sin\left(x_l(j-1/2)\pi\right) f_0(x_l).$$

Now, writing $\pi_j := (j - 1/2)\pi$, we have $\sin(x_l \pi_j) = \frac{1}{2i}(e^{ix_l\pi_j} - e^{-ix_l\pi_j})$ and thus

$$\left| \sum_{l=1}^{n} \sin\left(x_l(j-1/2)\pi\right) f_0(x_l) \right| \lesssim \left| n \int_0^1 \sin(t(j-1/2)\pi) f_0(t) dt \right| + n^{1-\alpha}$$

$$\leq \left| \frac{n}{2i} \left[ \int_0^1 e^{ix\pi_j} f_0(x) dx - \int_0^1 e^{-ix\pi_j} f_0(x) dx \right] \right| + n^{1-\alpha}$$

$$\lesssim n\pi_j^{-\alpha} + n^{1-\alpha} \asymp n\pi_j^{-\alpha},$$

for $f_0 \in C^\alpha([0,1])$, where the first inequality follows from Lemma A.4 below. We have thus shown that $|\mathbf{v}_j^T \mathbf{f}_0| \lesssim \sqrt{n}(j-1/2)^{-\alpha}$. Using this, that $\mu_j \asymp (n+1/2)c_n \frac{1}{(j-1/2)^2}$ by Lemma A.1 and that $\eta_j \lesssim 1$,

$$|b_m(x) - b_n(x)| \lesssim \sum_{j=m+1}^{n} \eta_j \mu_j v_j^i \sqrt{n}(j-1/2)^{-\alpha} \asymp nc_n \sum_{j=m+1}^{n} j^{-2-\alpha} \asymp nc_n m^{-(1+\alpha)}$$

as required. Next, we have,

$$t_m^2(x) = t_n^2(x) - \|\mathbf{r}_m(x)\|^2 = t_n^2(x) - \mathbf{k}_n(x)^T \sum_{j=m+1}^{n} \eta_j^2 \mathbf{v}_j \mathbf{v}_j^T \mathbf{k}_n(x).$$

Using again that $x = x_i$ and so $\mathbf{k}_n(x_i)^T \mathbf{v}_j = \mu_j v_j^i$, we have,

$$\mathbf{k}_n(x)^T \left( \sum_{j=m+1}^{n} \eta_j^2 \mathbf{v}_j \mathbf{v}_j^T \right) \mathbf{k}_n(x) = \sum_{j=m+1}^{n} \eta_j^2 \mu_j^2 (v_j^i)^2 \lesssim \frac{1}{n} \sum_{j=m+1}^{n} \frac{n^2 c_n^2}{(j-1/2)^4} \lesssim nc_n^2 m^{-3},$$

as required. Finally,

$$\sigma_m^2(x) = \sigma_n^2(x) + \mathbf{k}_n(x)^T \left( \sum_{j=m+1}^{n} \eta_j \mathbf{v}_j \mathbf{v}_j^T \right) \mathbf{k}_n(x), \tag{21}$$

and again at a design point $x = x_i$,

$$\mathbf{k}_n(x)^T \left( \sum_{j=m+1}^{n} \eta_j \mathbf{v}_j \mathbf{v}_j^T \right) \mathbf{k}_n(x) = \sum_{j=m+1}^{n} \eta_j \mu_j^2 (v_j^i)^2 \lesssim nc_n^2 m^{-3}.$$

This establishes the three relations at the design points $x = x_i$.

Turning to the general case $x \in [0,1]$, let $i$ be such that $x_i < x < x_{i+1}$. Recalling that the covariance function equals $k(x, x') = c_n(x \wedge x')$, we obtain

$$\mathbf{k}_n(x) = c_n \begin{bmatrix} x_1 \\ \vdots \\ x_i \\ x \\ \vdots \\ x \end{bmatrix} = \mathbf{k}_n(x_i) + c_n \begin{bmatrix} 0 \\ \vdots \\ 0 \\ x - x_i \\ \vdots \\ x - x_i \end{bmatrix},$$

where $0 < x - x_i < \frac{1}{n+1/2}$. Consider first the case of the SGPR variance $\sigma_m^2(x)$. The last display implies that $(\mathbf{k}_n(x) - \mathbf{k}_n(x_i))^T \mathbf{v}_j = c_n(x - x_i) \sum_{l=i+1}^{n} v_j^l$, and hence the quadratic term in (21) can be rewritten as

$$\mathbf{k}_n(x)^T \left( \sum_{j=m+1}^{n} \eta_j \mathbf{v}_j \mathbf{v}_j^T \right) \mathbf{k}_n(x) = \mathbf{k}_n(x_i)^T \left( \sum_{j=m+1}^{n} \eta_j \mathbf{v}_j \mathbf{v}_j^T \right) \mathbf{k}_n(x_i)$$

$$+ 2c_n \sum_{j=m+1}^{n} \eta_j \left[ (x - x_i) \sum_{l=i+1}^{n} v_j^l \right] \mathbf{v}_j^T \mathbf{k}_n(x_i) \tag{22}$$

$$+ c_n^2 \sum_{j=m+1}^{n} \eta_j \left[ (x - x_i) \sum_{l=i+1}^{n} v_j^l \right]^2.$$

The first term is exactly the quadratic term evaluated at the design point $x_i$, which we showed above has size $O(nc_n^2 m^{-3})$. Using Lemmas A.1 and A.4,

$$
\begin{aligned}
\sum_{l=i+1}^{n} v_j^l &= \frac{2}{\sqrt{2n+1}} \sum_{l=i+1}^{n} \sin(x_l(j-1/2)\pi) \\
&= \frac{2n}{\sqrt{2n+1}} \left( \int_{x_i}^{1} \sin(t(j-1/2)\pi)dt + O(n^{-1}) \right) \\
&= \frac{2n}{\sqrt{2n+1}} \frac{\cos(x_i(j-1/2)\pi)}{(j-1/2)\pi} + O(n^{-1/2}) = O\left( \frac{\sqrt{n}}{j-1/2} \right) + O(n^{-1/2}).
\end{aligned}
$$

By Lemma A.1, $v_j^T \mathbf{k}_n(x_i) = \mu_j v_j^i = O(c_n \sqrt{n} j^{-2})$. Since also $\eta_j \lesssim 1$, the second term in (22) is bounded by a multiple of

$$
\frac{c_n}{\sqrt{n}} \sum_{j=m+1}^{n} j^{-1} |\mathbf{v}_j^T \mathbf{k}_n(x_i)| \lesssim c_n^2 \sum_{j=m+1}^{n} j^{-3} \lesssim c_n^2 m^{-2} = O(nc_n^2 m^{-3}).
$$

The third term in (22) can similarly be shown to be $O(c_n^2 n^{-1} m^{-1}) = O(nc_n^2 m^{-3})$. We have thus shown that (22) is of size $O(nc_n^2 m^{-3})$ exactly as in the case where $x$ was a design point, which implies the desired bound for the SGPR variance $\sigma_m^2(x)$.

The case $t_m^2(x)$ follows in a similar fashion, simply replacing $\eta_j$ by $\eta_j^2$, while the bias $b_m(x)$ uses additionally the bound $|\mathbf{v}_j^T \mathbf{f}_0| \lesssim \sqrt{n}(j-1/2)^{-\alpha}$ derived above. In conclusion, the three quantities $b_m(x)$, $t_m^2(x)$ and $\sigma_m^2(x)$ satisfy the same bounds as when $x$ is a design point, which completes the proof. $\qquad \square$

The next standard bound controls the error when approximating a $C^\alpha$ function by its Riemann sum.

**Lemma A.4.** *If $f \in C^\alpha[0,1]$ with $\alpha \in (0,1]$, then*

$$
\left| \int_0^1 f(t)dt - \frac{1}{n} \sum_{i=1}^{n} f(x_i) \right| \le (n+1/2)^{-\alpha},
$$

*with $x_i = \frac{i}{n+1/2}$ for $i = 1, \dots, n$.*

The following lemma shows that for $m$ large enough, the bias, frequentist variance of the SGPR mean and SGPR variance can each be related to those of the full posterior derived in [29].

**Lemma A.5.** *If $m \gg n^{\frac{1}{1+2\gamma}\left(\frac{2+\alpha}{1+\alpha}\right)}$, then the SGPR satisfies*

$$
\begin{aligned}
b_m(x) &= b_n(x)(1 + o(1)) \\
t_m^2(x) &= t_n^2(x)(1 + o(1)) \\
\sigma_m^2(x) &= \sigma_n^2(x)(1 + o(1)).
\end{aligned}
$$

*Proof.* From [29], we have the following expressions for the full posterior:

$$
|b_n(x)| \lesssim n^{-\frac{\alpha}{1+2\gamma}}, \qquad t_n^2(x) \asymp n^{-\frac{2\gamma}{1+2\gamma}}, \qquad \sigma_n^2(x) \asymp n^{-\frac{2\gamma}{1+2\gamma}}.
$$

Thus, if we can show that the $O$ terms given in equations (18)-(20) are smaller than their respective terms above, we have the result. By Lemma A.3, we have $b_m(x) = b_n(x) + O(nc_n m^{-(1+\alpha)})$, and $nc_n m^{-(1+\alpha)} = n^{\frac{2}{1+2\gamma}} m^{-(1+\alpha)} \lesssim n^{-\frac{\alpha}{1+2\gamma}}$ if and only if $n^{\frac{1}{1+2\gamma}\left(\frac{2+\alpha}{1+\alpha}\right)} \ll m$. Thus, if $m \gg n^{\frac{1}{1+2\gamma}\left(\frac{2+\alpha}{1+\alpha}\right)}$ then $nc_n m^{-(1+\alpha)} = o(b_n(x))$ and $b_m(x) = b_n(x)(1 + o(1))$. Similarly,

$$
nc_n^2 m^{-3} \ll t_n^2(x) = \sigma_n^2(x) \asymp n^{-\frac{2\gamma}{1+2\gamma}} \Leftrightarrow m \gg n^{\frac{1}{1+2\gamma}}.
$$

Thus, $t_m^2(x) = t_n^2(x)(1 + o(1))$ and $\sigma_m^2(x) = \sigma_n^2(x)(1 + o(1))$ if $m \gg n^{\frac{1}{1+2\gamma}\left(\frac{2+\alpha}{1+\alpha}\right)}$ (in fact these are implied even at the smaller bound $n^{\frac{1}{1+2\gamma}}$). $\qquad \square$

*Proof of Theorem 3.5.* The proof follows from Proposition 3.2 combined with the two groups of expressions (7)-(9) and (18)-(20). $\qquad \square$

*Proof of Theorem 3.6.* The coverage equals
$$P_{f_0}(f_0(x) \in C_m^\delta) = P_{f_0}(|\nu_m(x) - f_0(x)| \le z_{1-\delta}\sigma_m(x)) = P_{f_0}(|V_m| \le z_{1-\delta}),$$
for $V_m \sim \mathcal{N}\left(\frac{b_m(x)}{\sigma_m(x)}, \frac{t_m^2(x)}{\sigma_m^2(x)}\right)$. Now, since $m \gg n^{\frac{1}{1+2\gamma}(\frac{2+\alpha}{1+\alpha})}$ we have $b_m(x) = b_n(x)(1 + o(1))$, $t_m^2(x) = t_n^2(x)(1 + o(1))$ and $\sigma_m^2(x) = \sigma_n^2(x)(1 + o(1))$ and thus,
$$\lim_{n\to\infty} P_{f_0}(|V_m| \le z_{1-\delta}) = \lim_{n\to\infty} P_{f_0}(|V_n| \le z_{1-\delta}) = \lim_{n\to\infty} P_{f_0}(f_0(x) \in C_n^\delta).$$
$\square$

**Lemma A.6.** *The Kullback-Leibler divergence between the SGPR $Q^*$ and the full posterior satisfies*
$$2KL(Q^* || \Pi(\cdot|Y)) = \mathbf{y}^T \left(\frac{1}{\sigma^2} \sum_{j=m+1}^{n} \frac{\mu_j}{\mu_j + \sigma^2} \mathbf{v}_j \mathbf{v}_j^T\right) \mathbf{y} + \sum_{j=m+1}^{n} \log \frac{\sigma^2}{\sigma^2 + \mu_j} + \frac{1}{\sigma^2} \sum_{j=m+1}^{n} \mu_j.$$
*Suppose further that $f_0 \in C^\alpha$, $\alpha \in (0,1]$, and consider the fixed design setting with the rescaled Brown motion prior of regularity $\gamma \in (0,1]$. If $m \gg n^{\frac{1}{1+2\gamma}}$, then*
$$E_{f_0} KL(Q^* || \Pi(\cdot|Y)) \asymp n^{\frac{2}{1+2\gamma}} m^{-1} + n^{\frac{3+2\gamma}{1+2\gamma}} m^{-1-2\alpha}.$$
*In particular, if $n^{\frac{1}{1+2\gamma}} \ll m \ll n^{\frac{2}{1+2\gamma}}$ then $E_{f_0} KL(Q^* || \Pi(\cdot|Y)) \to \infty$.*

*Proof.* We have from [31]
$$KL(Q || \Pi(\cdot|Y)) = \frac{1}{2}\left(\mathbf{y}^T\left(\mathbf{Q}_n^{-1} - \mathbf{K}_n^{-1}\right)\mathbf{y}\right) + \log \frac{|\mathbf{Q}_n|}{|\mathbf{K}_n|} + \frac{1}{\sigma^2} tr(\mathbf{K}_n - \mathbf{Q}_n),$$
where $\mathbf{K}_n = \mathbf{K}_{nn} + \sigma^2 I$, $\mathbf{Q}_n = Q_{nn} + \sigma^2 I$, $\mathbf{K}_{nn} = \sum_{j=1}^{n} \mu_j \mathbf{v}_j \mathbf{v}_j^T$ and $Q_{nn} = \sum_{j=1}^{m} \mu_j \mathbf{v}_j \mathbf{v}_j^T$. We thus have
$$\mathbf{Q}_n^{-1} - \mathbf{K}_n^{-1} = \sum_{j=m+1}^{n} \left(\frac{1}{\sigma^2} - \frac{1}{\sigma^2 + \mu_j}\right) \mathbf{v}_j \mathbf{v}_j^T = \frac{1}{\sigma^2} \sum_{j=m+1}^{n} \frac{\mu_j}{\sigma^2 + \mu_j} \mathbf{v}_j \mathbf{v}_j^T,$$
$\frac{|\mathbf{Q}_n|}{|\mathbf{K}_n|} = \prod_{j=m+1}^{n} \frac{\sigma^2}{\sigma^2 + \mu_j}$, and $tr(\mathbf{K}_n - \mathbf{Q}_n) = \sum_{j=m+1}^{n} \mu_j$, giving the results. We now consider the three terms given in the first part of the lemma. Defining $M := \sum_{j=m+1}^{n} \frac{\mu_j}{\mu_j + \sigma^2} \mathbf{v}_j \mathbf{v}_j^T$, we have
$$E_{f_0}\left[\mathbf{y}^T\left(\frac{1}{\sigma^2}\sum_{j=m+1}^{n} \frac{\mu_j}{\mu_j + \sigma^2} \mathbf{v}_j \mathbf{v}_j^T\right)\mathbf{y}\right] = \frac{1}{\sigma^2} E_{f_0}\left[(\mathbf{f}_0 + \boldsymbol{\varepsilon})^T M (\mathbf{f}_0 + \boldsymbol{\varepsilon})\right]$$
$$= \frac{1}{\sigma^2}\mathbf{f}_0^T M \mathbf{f}_0 + \frac{1}{\sigma^2} E_{f_0}\left[\boldsymbol{\varepsilon}^T M \boldsymbol{\varepsilon}\right].$$
Recalling from the proof of Lemma A.3 that $|\mathbf{v}_j^T \mathbf{f}_0| \lesssim \sqrt{n} j^{-\alpha}$, we can control
$$\frac{1}{\sigma^2}\mathbf{f}_0^T M \mathbf{f}_0 = \frac{1}{\sigma^2} \sum_{j=m+1}^{n} \frac{\mu_j}{\mu_j + \sigma^2}(\mathbf{v}_j^T \mathbf{f}_0)^2 \lesssim \frac{1}{\sigma^2} \sum_{j=m+1}^{n} n\mu_j j^{-2\alpha} \asymp n^{\frac{3+2\gamma}{1+2\gamma}} m^{-1-2\alpha}.$$
Next, we have
$$E_{f_0}[\boldsymbol{\varepsilon}^T M \boldsymbol{\varepsilon}] = E_{f_0} tr(\boldsymbol{\varepsilon}^T M \boldsymbol{\varepsilon}) = E_{f_0} tr(M\boldsymbol{\varepsilon}\boldsymbol{\varepsilon}^T) = tr(M E_{f_0}[\boldsymbol{\varepsilon}\boldsymbol{\varepsilon}^T]) = tr(M) \asymp n^{\frac{2}{1+2\gamma}} m^{-1}.$$
For the second term in the first part of the lemma,
$$\sum_{j=m+1}^{n} \log \frac{\sigma^2}{\sigma^2 + \mu_j} = -\sum_{j=m+1}^{n} \log\left(1 + \frac{\mu_j}{\sigma^2}\right) = -\sum_{j=m+1}^{n}\left(\frac{\mu_j}{\sigma^2} + O\left(\frac{\mu_j^2}{\sigma^4}\right)\right)$$
$$= -\frac{1}{\sigma^2}\sum_{j=m+1}^{n} \mu_j - O\left(\sum_{j=m+1}^{n} \frac{\mu_j^2}{\sigma^4}\right)$$
$$= -\frac{1}{\sigma^2} n^{\frac{2}{1+2\gamma}}/m - O(n^{\frac{4}{1+2\gamma}} m^{-3}).$$
Hence, adding these expressions to the third term in the first part of the lemma yields
$$E_{f_0} KL(Q || \Pi(\cdot|Y)) \asymp n^{\frac{2}{1+2\gamma}} m^{-1} + n^{\frac{3+2\gamma}{1+2\gamma}} m^{-1-2\alpha} - O(n^{\frac{4}{1+2\gamma}} m^{-3}).$$
The last term is of strictly smaller order than the first two if $m \gg n^{\frac{1}{1+2\gamma}}$.
$\square$

## B    Additional numerical simulations

We provide here additional simulations to those in Section 5, investigating further the minimal number of inducing variables needed for good pointwise UQ and the sensitivity of the results to misspecification of the noise distribution. To illustrate the pointwise approach to UQ using full posterior inference and SGPRs, one can refer to Figure 1 in the paper.

**Threshold for the number of inducing variables.** We showed in Sections 3 and 5 that for $m \gg n^{\frac{1}{1+2\gamma}(\frac{2+\alpha}{1+\alpha})}$ inducing variables, the SGPR behaves similarly to the true posterior for pointwise inference. When $\alpha = \gamma$, the threshold for minimax convergence rates for *estimation* in $L^2$ is the smaller bound $m \gg n^{\frac{1}{1+2\alpha}}$ [22]. We next investigate how the SGPR behaves near this threshold.

Return to the fixed ($X_i = i/(n+1/2)$) and uniform random ($X_i \sim^{iid} U[0,1]$) design settings on $[0,1]$ and consider the same Gaussian processes as before: rescaled Brownian motion, Matérn and rescaled square exponential. We consider three different values of $m$ given by $m^*_- = n^{\frac{1}{1+2\gamma}}/\log n$, $m^*_+ = n^{\frac{1}{1+2\gamma}} \cdot \log n$, so that these values of $m$ are just below and above the threshold $n^{\frac{1}{1+2\gamma}}$, respectively, and the full posterior case $m = n$. We consider the $1-$Hölder function $f_0(x) = |x-1/2|$ and $2-$Hölder function $f_0(x) = \text{sign}(x-1/2)|x-1/2|^2$.

| GP | Coverage | | | Length | | | RMSE | | | NLPD | | |
|---|---|---|---|---|---|---|---|---|---|---|---|---|
| | $m^*_-$ | $m^*_+$ | $n$ | $m^*_-$ | $m^*_+$ | $n$ | $m^*_-$ | $m^*_+$ | $n$ | $m^*_-$ | $m^*_+$ | $n$ |
| **Fixed Design:** | $n = 1000, (\alpha, \gamma) = (1, 0.5)$ | | | | | | | | | | | |
| rBM | 1.00 | 0.98 | 0.98 | 0.52 | 0.42 | 0.42 | 0.06 | 0.09 | 0.09 | -0.85 | -0.89 | -0.89 |
| Matérn | 1.00 | 0.98 | 0.98 | 0.69 | 0.50 | 0.50 | 0.07 | 0.11 | 0.11 | -0.59 | -0.70 | -0.70 |
| SE | 1.00 | 0.93 | 0.93 | 2.73 | 0.66 | 0.66 | 0.08 | 0.18 | 0.18 | 0.74 | -0.28 | -0.28 |
| **Fixed Design:** | $n = 1000, (\alpha, \gamma) = (2, 1.5)$ | | | | | | | | | | | |
| rBM | 0.98 | 0.98 | 0.98 | 0.18 | 0.17 | 0.17 | 0.04 | 0.04 | 0.04 | -1.72 | -1.74 | -1.74 |
| Matérn | 1.00 | 0.95 | 0.95 | 0.56 | 0.22 | 0.22 | 0.04 | 0.05 | 0.05 | -0.82 | -1.51 | -1.51 |
| SE | 1.00 | 0.91 | 0.91 | 2.29 | 0.31 | 0.31 | 0.05 | 0.09 | 0.09 | 0.56 | -0.99 | -0.99 |
| **Random Design:** | $n = 500, (\alpha, \gamma) = (1, 0.5)$ | | | | | | | | | | | |
| rBM | 1.00 | 0.97 | 0.97 | 0.59 | 0.50 | 0.50 | 0.08 | 0.10 | 0.10 | -0.70 | -0.75 | -0.75 |
| Matérn | 1.00 | 0.96 | 0.96 | 0.87 | 0.59 | 0.59 | 0.07 | 0.12 | 0.12 | -0.38 | -0.58 | -0.58 |
| SE | 1.00 | 0.94 | 0.94 | 2.70 | 0.77 | 0.77 | 0.10 | 0.21 | 0.21 | 0.73 | -0.13 | -0.13 |
| **Random Design:** | $n = 500, (\alpha, \gamma) = (2, 1.5)$ | | | | | | | | | | | |
| rBM | 0.98 | 0.95 | 0.95 | 0.23 | 0.23 | 0.23 | 0.05 | 0.05 | 0.05 | -1.49 | -1.49 | -1.49 |
| Matérn | 1.00 | 0.96 | 0.96 | 0.57 | 0.29 | 0.29 | 0.05 | 0.07 | 0.07 | -0.79 | -1.19 | -1.19 |
| SE | 1.00 | 0.95 | 0.95 | 2.10 | 0.39 | 0.39 | 0.05 | 0.10 | 0.10 | 0.47 | -0.86 | -0.86 |

Table 3: Comparison of SGPR and full posterior for 90% pointwise credible intervals for different values of $(n, \alpha, \gamma)$ in dimension $d = 1$. The column headers describe how many inducing variables were used, with $m^*_- := n^{\frac{1}{1+2\gamma}}/\log(n)$ and $m^*_+ := n^{\frac{1}{1+2\gamma}} \cdot \log(n)$ and the full posterior represented by $n$. The hypothesised bound $n^{\frac{1}{1+2\gamma}}$ equals 32, 6, 22 and 5 in the four different regimes (top-to-bottom).

The results of our experiments for different values of $(n, \alpha, \gamma)$ are given in Table 3; we note here that we exclude estimates of the standard error in this Table due to space constraints, but they are comparable to those presented in Table 1 in the main text. In all cases, one can see that the coverage, length and RMSE are practically indistinguishable between the cases $m = m^*_+$ and the full posterior $m = n$, suggesting that the choice of $m \gg n^{\frac{1}{1+2\gamma}}$ inducing variables is sufficient to achieve the same performance as the full posterior for pointwise inference. Furthermore, for fixed design, we again see the Brownian motion with both choices $m = m^*_+$ and $m = n$ achieves the expected coverage of 0.98 predicted by our theory.

Conversely, for $m = m^*_-$ inducing variables, the coverage and credible interval length was larger in almost all cases, sometimes significantly so. This suggests that below this threshold, the SGPR is no longer a good approximation for the true posterior, matching the findings in [22, 23] for $L^2$-type inference. On the other hand, the RMSE is usually reduced, indicating that additional smoothing helps the sparse posterior mean for estimation, which is not surprising since in our simulation choices

the prior undersmooths the truth. The main message is that taking fewer inducing points than this threshold leads to a significant increase in the credible interval width, making the UQ less informative and overly conservative. It is interesting that this threshold shows so clearly, since $m^*_-$ and $m^*_+$ differ only by a subpolynomial factor.

The same story holds for random design, with some additional fluctuations due to this additional source of randomness. We recall that for fixed design, the credible intervals have deterministic length, whereas for random design these are also now random.

It would be interesting to extend our theoretical results to the threshold $n^{\frac{1}{1+2\gamma}}$ as suggested by these simulations. However, the mismatch between pointwise loss and the $L^2$-distance induced by the likelihood makes using standard tools from Bayesian nonparametrics difficult, see [15] for further discussion.

**Multidimensional setting.** Consider again $d = 10$ dimensional covariates with the Matérn kernel of smoothness 1/2, and the corresponding threshold $n^{\frac{d}{d+2\gamma}}$ needed for the minimax contraction rate in $L^2$ [22]. We take uniform random design $X_i \sim^{iid} \mathcal{U}([-1/2, 1/2]^d)$, true function $f_0(x) = \|x\|^\alpha$ for $\alpha = 0.9$ and investigate the pointwise $90\%-$credible intervals at the point $x_0 = (0, \ldots, 0)$. We now take three values of $m$ given by $m^*_{d-} = n^{\frac{d}{d+2\gamma}} / \log n$, $m^*_{d+} = n^{\frac{d}{d+2\gamma}} \cdot \log n$ and $m = n$. The results are presented in Table 4. Indeed, one sees that the variational posterior performs identically to the full posterior once at least $m^*_{d+} = n^{d/(d+2\gamma)} \cdot \log n \gg n^{\frac{d}{d+2\gamma}}$ inducing variables have been used, while $m^*_{d-}$ inducing variables do not seem to be enough to get the same performance.

| Coverage | | | Length | | | RMSE | | | NLPD | | |
|---|---|---|---|---|---|---|---|---|---|---|---|
| $m^*_{d-}$ | $m^*_{d+}$ | $n$ | $m^*_{d-}$ | $m^*_{d+}$ | $n$ | $m^*_{d-}$ | $m^*_{d+}$ | $n$ | $m^*_{d-}$ | $m^*_{d+}$ | $n$ |
| 0.93 | 1.00 | 1.00 | 2.59 | 2.40 | 2.40 | 0.91 | 0.73 | 0.73 | 1.35 | 1.10 | 1.10 |

Table 4: Comparison of SGPR and full posterior with Matérn prior for 90% pointwise credible intervals for $(d, \alpha, \gamma) = (10, 0.9, 0.5)$. The column headers describe how many inducing variables were used with $m^*_{d-} = n^{\frac{d}{d+2\gamma}} / \log(n)$ and $m^*_{d+} = n^{\frac{d}{d+2\gamma}} \cdot \log(n)$ and the full posterior represented by $n$. Here, $n^{\frac{d}{d+2\gamma}} = 284$.

**Misspecified noise.** We now take the same fixed design setting as in Section 3 of the main paper, but sample $Y_i = f_0(X_i) + \varepsilon_i$, where $\varepsilon_i$ are independent standard Laplace random variables. Based on Table 5, the results are much the same as in the well-specified case, so that the credible sets demonstrate some robustness to noise misspecification. Note that while the coverage remains broadly similar, the credible intervals are slightly wider and have larger RMSE reflecting the larger noise.

| | Smoothness | | Coverage | | Length | | RMSE | | NLPD | |
|---|---|---|---|---|---|---|---|---|---|---|
| GP | $\alpha$ | $\gamma$ | SGPR | GP | SGPR | GP | SGPR | GP | SGPR | GP |
| rBM | 1 | 0.5 | 0.98 | 0.98 | 0.49 | 0.49 | 0.11 | 0.11 | -0.71 (0.15) | -0.71 (0.15) |
| Matérn | 1 | 0.5 | 0.98 | 0.98 | 0.58 | 0.58 | 0.12 | 0.12 | -0.58 (0.17) | -0.58 (0.17) |
| rBM | 0.5 | 0.5 | 0.69 | 0.69 | 0.49 | 0.49 | 0.22 | 0.22 | 0.11 (0.32) | 0.11 (0.32) |
| Matérn | 0.5 | 0.5 | 0.88 | 0.88 | 0.58 | 0.58 | 0.20 | 0.20 | -0.17 (0.20) | -0.17 (0.20) |

Table 5: Misspecified noise. SGPR summary statistics for 90% pointwise credible intervals for different values of $(\alpha, \gamma)$ with regular fixed design on $[0, 1]$ with $n = 1000$ and Laplace noise. The SGPR uses $m^* = n^{\frac{1}{1+2\gamma} \frac{2+\alpha}{1+\alpha}}$ inducing variables, which is 126 and 316 in the two regimes.

### B.1 Definitions of simulation statistics

For completeness, we define here the statistics reported in the simulation sections based on $M$ Monte-Carlo samples. For $Y_i^{(j)} = f_0(X_i) + \varepsilon_i^{(j)}$, $i = 1, \ldots, n$ and $j = 1, \ldots, M$, let $C_m^{(j)}(x_0)$ denote the credible interval for $f(x_0)$, $\bar{f}_m^{(j)}(x_0)$ and $\sigma_m^{(j)}(x_0)^2$ the posterior mean and variance from the SGPR with $m$ inducing variables computed from the $j^{th}$ realisation of the data. We then consider the Monte-Carlo estimates of the *coverage*, credible interval *length*, *root-mean square error (RMSE)*

and *negative log-predictive density* (NLPD) given by

$$\widehat{\text{Cov}} = \frac{1}{M} \sum_{j=1}^{M} \mathbf{1}\{f_0(x_0) \in C_m^{(j)}(x_0)\}, \quad \widehat{RMSE} = \sqrt{\frac{1}{M} \sum_{j=1}^{M} (\bar{f}_m^{(j)}(x_0) - f_0(x_0))^2},$$

$$\widehat{\text{Len}} = \frac{1}{M} \sum_{j=1}^{M} \text{Length}\left(C_m^{(j)}(x_0)\right), \quad \widehat{NLPD} = -\frac{1}{M} \sum_{j=1}^{M} \log \mathcal{N}\left(f_0(x_0) | f_m^{(j)}(x_0), \sigma_m^{(j)}(x_0)^2\right),$$

respectively, where $\mathcal{N}\left(f_0(x_0) | f_m^{(j)}(x_0), \sigma_m^{(j)}(x_0)^2\right)$ denotes the density of the Gaussian distribution with mean $f_m^{(j)}(x_0)$ and variance $\sigma_m^{(j)}(x_0)^2$ evaluated at the point $f_0(x_0)$.

