# OpenReview forum: "Pointwise uncertainty quantification for sparse variational Gaussian process regression with a Brownian motion prior"
_NeurIPS.cc/2023/Conference — NeurIPS 2023 poster_

### Official Review · Reviewer_oD8s · 2023-06-15

**Soundness:** 3 good
**Presentation:** 2 fair
**Contribution:** 2 fair
**Rating:** 6
**Confidence:** 3

**Summary:**

Broadly, the paper addresses the issue of the quality of the approximate posterior of a sparse GP model in terms of uncertainty qualification (UQ). The paper chooses the setup with eigenvector-inducing variables and rescaled Brownian motion prior. However, as shown, the results also extend to other kernels (squared exponential (RBF) and Matérn kernels). Precisely, the paper's main conclusion is that with a well-calibrated prior and a sufficiently large number of inducing variables, the uncertainty quantification (UQ) obtained is reliable, though it can be conservative. The paper presents theoretical results on the rate and quality of SVGP posterior, nature of credible sets, and contraction rate of approximate posterior. The theoretical results are shown to match the empirical results based on experiments on simulation data and semi-simulation data.

**Strengths:**

The topic of the paper is relevant as Gaussian processes are go-to models for many tasks, including Bayesian optimization, active learning, and various sequential tasks. Uncertainty quantification is one of the crucial properties of the Gaussian process models that make them popular for these tasks. Recently, a lot of research has been done on the quality of the approximate posterior of sparse GP models, and this paper rightly fits in there in terms of evaluating the quality of approximate posterior in terms of uncertainty quantification. The claims of the paper are sensible, and the detailed derivations in the appendix also look logical (I would like to see what other reviewers think about the proofs as that is not my core area). Apart from a few places, the paper's notations and flow are good and convenient to follow. The experiments are also chosen wisely and demonstrate the theoretical results.

**Weaknesses:**

A couple of weaknesses that I find in this paper are:

* The paper talks about sparse variational Gaussian process models (SVGP) throughout, but actually, the model considered in the paper is sparse Gaussian process regression (SGPR). It can be clearly seen from Eq (3). It is better to make this clear in the paper. With a non-Gaussian likelihood, the equations for the posterior are different from those presented here.
* Section 2.1 has issues with the notations.
  - As pointed out earlier, the equations are of SGPR and not SVGP.
  - L104 mean of the GP `m` is a typo?
  - L117-L118 starts writing cov. Should it be `k`?
  - $K$ is the gram matrix and should be bold as I believe the paper tries to follow that notation by making $k_n(x)$ bold.
* A key takeaway message of the paper is missing. There are a bunch of theorems, derivations, and contributions, but I am still trying to figure out the main takeaway message from the paper that I should remember next time I plan to use a sparse GP model.
* As we discuss uncertainty quantification, does it make sense to report negative log predictive density (NLPD) in the experiments along with the RMSE?
* A plot of the true posterior, approximate posterior, and credible set would be good to demonstrate the setup and the output.
* Theorem 3.5 derives a bound for the number of inducing variables `m`. How does it relate to Burt et al. (2019), as they also derive a bound for the number of inducing variables?


**Questions:**

* Is the metric for the quality of the approximate posterior different from the metric of UQ? Is having a “good” quality approximate posterior but not a well-calibrated UQ possible? As the authors also point out in the paper, there has been a lot of recent research evaluating the quality of approximate posterior and convergence rates. Is it much different?
* What are the connections with Burt et al. (2019), and where do the authors agree and disagree with them? In general, I believe there should be more connections drawn with them.
* A minor comment, references to the appendix need to be included. I do not see any reference to the appendix in the main paper, so connecting derivations and their corresponding sections becomes tricky.
* More practical question: how to select the value of $\alpha, \gamma$ for a real-world dataset, let us say a UCI data set. I believe more evaluations should be done by extending section 4. It would make the work more applicable, in my opinion.
* More questions are written in the weakness section.


**Limitations:**

The authors do not discuss them, and I do not see a limitation directly of this work.

---

> ### Author Rebuttal · Authors · 2023-08-07
>
> We thank the reviewer for their constructive suggestions and helpful comments. Our response:
>
> - The paper talks about sparse variational Gaussian process models (SVGP) throughout, but actually, the model considered in the paper is sparse Gaussian process regression (SGPR).
>
> We have changed our terminology to SGPR as per the reviewer's suggestion.
>
> - Notational issues.
>
> We have fixed all of these, thank you.
>
> - A key takeaway message of the paper is missing. There are a bunch of theorems, derivations, and contributions, but I am still trying to figure out the main takeaway message from the paper that I should remember next time I plan to use a sparse GP model.
>
> We have tried to clearly summarize these via bullet points in the introduction. In summary, we give some first ideas of the various regimes one can expect for UQ using SGPR (reliable but conservative UQ in some cases, bad UQ in others). The practical messages also seem to apply in practice to the widely used Matern kernel. Please see our general response for more details.
>
> - As we discuss uncertainty quantification, does it make sense to report negative log predictive density (NLPD) in the experiments along with the RMSE?
>
> We have added this to the simulations, thank you.
>
> - A plot of the true posterior, approximate posterior, and credible set would be good to demonstrate the setup and the output.
>
> This is a very good idea. We would have liked to add this to the main text, but were unable due to space constraints. We have hence added this image in Section 8 of the supplement (see the figure attached with our general author response).
>
> - Theorem 3.5 derives a bound for the number of inducing variables m. How does it relate to Burt et al. (2019), as they also derive a bound for the number of inducing variables?
>
> Please see our general response.
>
> - Is the metric for the quality of the approximate posterior different from the metric of UQ? Is having a “good” quality approximate posterior but not a well-calibrated UQ possible? As the authors also point out in the paper, there has been a lot of recent research evaluating the quality of approximate posterior and convergence rates. Is it much different?
>
> Yes, these are fundamentally different questions, please see our general reponse ("Connection with Burt et al. 2019"). If the true posterior is bad for UQ, a good quality approximation will also be bad. One can also find examples (outside the GP setting) where a variational posterior performs better than the true posterior, e.g. Section 5.2 of Zhang & Gao (Annals of Statistics 2020).
>
> - What are the connections with Burt et al. (2019), and where do the authors agree and disagree with them? In general, I believe there should be more connections drawn with them.
>
> Please see our general response. We have added some brief comments in the text, but these were sadly limited due to space constraints.
>
> - A minor comment, references to the appendix need to be included. I do not see any reference to the appendix in the main paper, so connecting derivations and their corresponding sections becomes tricky.
>
> We placed all proofs in Section 7, as indicated in the 'Organization' paragraph on p3. Since all proofs are in the supplement and the section is not too long, we prefer not to add specific references after each result.
>
> - More practical question: how to select the value of $\alpha,\gamma$ for a real-world dataset, let us say a UCI data set. I believe more evaluations should be done by extending section 4. It would make the work more applicable, in my opinion.
>
> The natural approach would be to select $\gamma$ by maximizing the marginal likelihood or evidence lower bound (e.g. how the lengthscale is usually picked for the Matern). We have run some simulations in this way, which seem to match the messages we derive in this paper. So our results should extend to adaptive procedures for 'nice' truths.
>
> However, UQ theory for adaptive procedures is a subtle topic, which typically requires further assumptions (e.g. self-similarity) on the true generative function to even be possible. There exist functions that can trick an adaptive procedure into thinking they are smoother than they really are, and consequently make the resulting UQ procedure overconfident (= bad/misleading), see e.g. the discussion paper of Szabo, van der Vaart and van Zanten (AOS 2015). Given such functions will also cause the adaptive SGPR to fail here (like any adaptive procedure), we prefer not to include partial simulations in this manuscript that may not give the full picture of what happens.
>
> Regarding theory, we plan to extend our results to adaptation (i.e. picking the smoothness parameter in a data-driven way) in future. However, it is typical to first consider the non-adaptive case both for simplicity and also to understand the effect of the prior smoothness choice on the results. Indeed all the SVGPR theory papers we have cited (Burt et al ICML 2019, Nieman et al. JMLR 2021, Nieman et al. arXiv 2022, Vakili et al. ICML 2022) are non-adaptive.
>
> Extending our results to the adaptive setting will require significant technical and conceptual work, which is beyond the scope of our manuscript. In particular, one might hope that taking fewer inducing variables might counteract the overconfidence in such settings when faced with bad truths.
>
> Regarding $\alpha$, this is the smoothness of the unknown true function but is not needed for the method ($\gamma$ plays this role and can be picked as described above). While our bound for the number of inducing variables required does depend on $\alpha$, one can use a weaker but still meaningful bound. Since $\alpha \mapsto \frac{2+\alpha}{1+\alpha}$ is decreasing on $(0,\infty)$, our bound satisfies $n^{\frac{1}{1+2\gamma}\frac{2+\alpha}{1+\alpha}} \ll n^\frac{2}{1+2\gamma}$, so that it is enough to take at least this many inducing variables, which only depends on $\gamma$. Note that for $\gamma>1/2$, this will be less than linear in $n$.

---

> > ### Comment · Reviewer_oD8s · 2023-08-11
> >
> > Thank you for the response!
> >
> > > As we discuss uncertainty quantification, does it make sense to report negative log predictive density (NLPD) in the experiments along with the RMSE?
> > We have added this to the simulations, thank you.
> >
> > Is it possible to share the numbers/table with RMSE and NLPD here?

---

> > > ### Author Response · Authors · 2023-08-12
> > > **NLPD Values**
> > >
> > > Hello!
> > >
> > > Unfortunately I wasn’t able to get the latex tables to compile here, but here are the values for the NLPD corresponding to the cases presented in the original version (of course in our updated document we include also the coverage, length and RMSE, but doing so here makes it quite difficult to read. Note here that the column SGPR represents the NLPD for the variational posterior, and GP the NLPD for the normal posterior.
> > >
> > > ### Table 1
> > >
> > > Prior   | alpha | gamma | SGPR  | GP
> > >
> > > —————————— ——————
> > >
> > > **Fixed Design**
> > >
> > > —————————— ——————
> > >
> > > rBM    | 1.0     | 0.5        | -0.90   | -0.90
> > >
> > > Mat    | 1.0     | 0.5        | -0.68   | -0.68
> > >
> > > SE      | 1.0     | 0.5        | -0.67   | -0.67
> > >
> > > —————————— ——————
> > >
> > > rBM    | 0.5   | 0.5        | -0.09   | -0.09
> > >
> > > Mat    | 0.5    | 0.5       | -0.40   | -0.40
> > >
> > > SE      | 0.5    | 0.5       | -0.21   | -0.21
> > >
> > > —————————— ——————
> > >
> > > **Random Design**
> > >
> > > —————————— ——————
> > >
> > > rBM    | 1.0     | 0.5      | -0.65   | -0.65
> > >
> > > Mat    | 1.0     | 0.5       | -0.55   | -0.55
> > >
> > > SE      | 1.0     | 0.5       | -0.02   | -0.02
> > >
> > > —————————— ——————
> > >
> > > rBM    | 0.3     | 0.5      | 2.23    | 2.23
> > >
> > > Mat    | 0.3     | 0.5       | 0.91   | 0.91
> > >
> > > SE      | 0.3     | 0.5       | 0.36   | 0.36
> > >
> > > —————————— ——————
> > >
> > >  ### Table 2
> > >
> > > **Multidimensional Random Design**
> > >
> > > Type | rho |SGPR | GP
> > >
> > > —————————— ——————
> > >
> > > Uniform    | n/a | 0.96 | 0.96
> > >
> > > Gaussian | 0.0 |  1.34 | 1.34
> > >
> > > Gaussian | 0.2 |  1.90    | 1.90
> > >
> > > Gaussian | 0.5 |  1.36    | 1.36
> > >
> > > —————————— ——————
> > >
> > > **Semi-synthetic Data**
> > >
> > > —————————— ——————
> > >
> > > SGPR | GP
> > >
> > > 3.22   | 3.22
> > >
> > >
> > > Thanks.

---

> > > > ### Comment · Reviewer_oD8s · 2023-08-14
> > > >
> > > > Thanks for sharing the table. I am raising my score!
> > > >
> > > > I believe a discussion on extending the method to more priors (RBF, Matern, etc.) and an elaborate section on connection with Burt *et al.* (2019) would strengthen the work. Also, while reporting the numbers NLPD or RMSE, kindly report the standard deviation as well by performing K-fold cross-validation.

---

### Official Review · Reviewer_sbVc · 2023-07-03

**Soundness:** 4 excellent
**Presentation:** 3 good
**Contribution:** 3 good
**Rating:** 7
**Confidence:** 3

**Summary:**

The paper gives some theoretical results for frequentist coverage of the credible sets for sparce variational Gaussian processes using a class of Brownian motion based priors. It provides some conclusions relating the smoothness of the prior vs the smoothness of the true underlying function and the resulting ability of the Bayesian inference to produse reliable frequentist uncertainty estimates.

**Strengths:**

The paper is very well written and easy to follow at a high level. The theoretical results are summarised well and give a good indication of the overall contribution of the paper.

The contribution seems to be an important starting point for providing guarantees for pointwise uncertainty estimates in SVGPs. It gives enough details to indicate the exact extent to which the results are valid.

The theoretical results are provided with a sufficient level of detail to be accessible by a more general (less statistically inclined) audience.

The discussion provides a number of directions for future work that can be taken up by the community.

**Weaknesses:**

The main motivation for the work is the wide use of SVGPs in practice. However, the results in the paper consider only very rough (BM) priors which are rarely used in practice, and only touch on smoother, more commonly used options such as Matern kernels. Admittedly, it is still an important contribution that further work can build on.

**Questions:**

It would be useful for my own understanding if the authors commented more on the statement that $L_2$-type credible sets are less reflective of actual practice than pointwise credible sets.

I would also be curious why the authors chose this venue for publishing this work, given the theoretical nature of the results.

**Limitations:**

It is possible that the paper will not be very accessible to the broader audience that it is addressing due to the theoretical nature of the main results. That said, I think the authors did a good job highlighting the exact contribution in the introduction and it should be accessible for practitioners in informing them of the possible consequences in the estimates of credible intervals based on the choice of priors. As the choice of prior is typically difficult in practice, these types of results are valuable in practice.

---

> ### Author Rebuttal · Authors · 2023-08-07
>
> We thank the reviewer for their constructive suggestions and helpful comments. Our response:
>
> - The main motivation for the work is the wide use of SVGPs in practice. However, the results in the paper consider only very rough (BM) priors which are rarely used in practice, and only touch on smoother, more commonly used options such as Matern kernels. Admittedly, it is still an important contribution that further work can build on.
>
> We fully agree it would be better to derive results for the Matern or squared exponential, but we do not currently have the tools to do so. We compromised by studying a more tractable prior sharing many similarities with the Matern. This allowed us to derive precise theoretical results, whose main messages seem to also apply to the Matern, as we verify numerically. Please see our general reponse for more details.
>
> - It would be useful for my own understanding if the authors commented more on the statement that $L_2$-type credible sets are less reflective of actual practice than pointwise credible sets.
>
> Many theoretical results consider $L_2$-credible sets for the entire function $f$ of the form $C_n = \\{ f: ||f-\hat{f}||_{L_2} \leq q_n \\}$ with $\Pi(C_n|D_n) = 0.95$. The structure of theses $L_2$-sets interacts nicely with Gaussian processes, since a GP can be naturally expanded in terms of an $L_2$ basis via its Karhunen-Loeve expansion. This allows an easier mathematical analysis. However, such credible sets are not so intuitive in practice, e.g. they contain unbounded functions and one cannot actually plot an $L_2$-ball. Indeed, two functions can be extremely close in $L_2$, but arbitrarily far apart at any given point $x_0$, which does not match the more practical intuition of functions being 'close'. The more usual approach in practice is to plot credible bands about the posterior mean, which is a pointwise concept (see the figure in the pdf file attached to the general author response). Pointwise UQ also has the interpretation of measuring the uncertainty of a prediction $f(x_0)$ for a given input $x_0$, which the global $L_2$ set does not.
>
> - I would also be curious why the authors chose this venue for publishing this work, given the theoretical nature of the results.
>
> Sparse variational GPs are widely used in the ML community, but much less so in the traditional statistics community. Hence we chose to submit to an ML venue to try and reach the community actually working with such methods. Given the length of our work, we felt an ML conference would be an appropriate venue for our submission.
>
> - It is possible that the paper will not be very accessible to the broader audience that it is addressing due to the theoretical nature of the main results. That said, I think the authors did a good job highlighting the exact contribution in the introduction and it should be accessible for practitioners in informing them of the possible consequences in the estimates of credible intervals based on the choice of priors. As the choice of prior is typically difficult in practice, these types of results are valuable in practice.
>
> Thank you for positive comments. We have indeed tried to clearly summarize our main messages, which is always a challenge for theoretical results.

---

> > ### Comment · Reviewer_sbVc · 2023-08-18
> >
> > Thank you for the response and clarification.

---

### Official Review · Reviewer_BMyX · 2023-07-03

**Soundness:** 3 good
**Presentation:** 3 good
**Contribution:** 3 good
**Rating:** 6
**Confidence:** 3

**Summary:**


The paper gives algorithms for point wise estimation and uncertainty quantification (error bars) for
sparse variational Gaussian processes with the Brownian motion as prior. The SGVP model though widely used in practice, lacks theoretical guarantees and this paper is the first to establish error guarantees for the model with Brownian priors. Experiments are presented to validate the results. The priors for SVGP models used in practice are not Brownian, extending the results in this paper to these priors remains an open question.


**Strengths:**

- This paper is among the first to address rigorously the statistical inference properties of the low rank approximations used in Gaussian processes.

The main result establishes rigorous bounds on the size of the confidence interval for the SGVP prediction given bounds on the smoothness of the prior and the ground truth and the number of eigenvalues retained in the approximation of the kernel matrix. This is a new and useful result for SGVPs.

- The analysis is particular to the case of Brownian motion priors. BM priors have eigenvectors that can be expressed in closed form (they correspond to the trigonometric functions) and have strong self-similarity properties. However, in the experiments it is observed that the qualitative behavior that is observed for the BM priors is also observed for the Matern and squared exponential kernels used in practice.


**Weaknesses:**


- The result does not extend in a straightforward way to the Matern and squared exponential kernels used in practice with SGVPs. Brownian priors have many special properties that are not shared with other Gaussian processes used in ML.

- The result although a useful addition to the literature on SGVPs do not seem not be of enough general interest for a clear accept
and need further development for analyzing the instances of SGVPs used in practice.

- The results are non adaptive as they require a pre-specified prior smoothness which is not a realistic assumption in practice.



**Questions:**


Pg 2, the wedge notation for the minimum, the definitions of smoothness and rate should be introduced before stating main results.

It may be helpful to look at the stochastic process literature where the various properties of BM were originally developed, there may be other stochastic processes like the Ornstein-Uhlenbeck process considered in the stochastic process literature whose spectra are similar to the BM, however these do not seem to have not been used for SGVP.



**Limitations:**

Limitations are discussed clearly by the authors.

---

> ### Author Rebuttal · Authors · 2023-08-07
>
> We thank the reviewer for their constructive suggestions and helpful comments. Our response:
>
> - The result does not extend in a straightforward way to the Matern and squared exponential kernels used in practice with SGVPs. Brownian priors have many special properties that are not shared with other Gaussian processes used in ML.
>
> We fully agree it would be better to derive results for the Matern or squared exponential, but we do not currently have the tools to do so. We compromised by studying a more tractable prior sharing many similarities with the Matern. This allowed us to derive precise theoretical results, whose main messages seem to also apply to the Matern, as we verify numerically. Please see our general reponse for more details.
>
> - The results are non adaptive as they require a pre-specified prior smoothness which is not a realistic assumption in practice.
>
> We fully agree and plan to extend our results to adaptation in future. However, it is typical to first consider the non-adaptive case both for simplicity and also to understand the effect of the prior smoothness choice on the results. Indeed all the SVGPR theory papers we have cited (Burt et al ICML 2019, Nieman et al. JMLR 2021, Nieman et al. arXiv 2022, Vakili et al. ICML 2022) are non-adaptive.
>
> The natural approach would be to select the parameter $\gamma$ by maximizing the marginal likelihood or evidence lower bound (e.g. how the lengthscale is usually picked for the Matern). We have run some simulations in this way, which seem to match the messages we derive in this paper. So our results should extend to adaptive procedures for 'nice' truths.
>
>  However, UQ theory for adaptive procedures is a subtle topic, which typically requires further assumptions (e.g. self-similarity) on the true generative function to even be possible. There exist functions that can trick an adaptive procedure into thinking they are smoother than they really are, and consequently make the resulting UQ procedure overconfident (= bad/misleading), see e.g. the discussion paper of Szabo, van der Vaart and van Zanten (AOS 2015). Given such functions will also cause the adaptive SGPR to fail here (like any adaptive procedure), we prefer not to include partial simulations in this manuscript that may not give the full picture of what happens.
>
> Extending our results to the adaptive setting will require significant technical and conceptual work, which is beyond the scope of our manuscript. In particular, one might hope that taking fewer inducing variables might counteract the overconfidence in such settings when faced with bad truths.
>
> - Pg 2, the wedge notation for the minimum, the definitions of smoothness and rate should be introduced before stating
> main results.
>
> We changed $\wedge$ to $\min$ in the introduction. We prefer not to introduce the rather lengthy definition of Holder smoothness and rate here since we feel this would break the reader's flow in the introduction.
>
> - It may be helpful to look at the stochastic process literature where the various properties of BM were originally developed, there may be other stochastic processes like the Ornstein-Uhlenbeck process considered in the stochastic process literature whose spectra are similar to the BM, however these do not seem to have not been used for SGVP.
>
> Thank you for this is an interesting point. We will have a look to see how far these results can be extended, but have not had a chance to do so yet.

---

> > ### Comment · Reviewer_BMyX · 2023-08-13
> >
> > Thanks for your response!

---

### Official Review · Reviewer_tkgm · 2023-07-05

**Soundness:** 3 good
**Presentation:** 2 fair
**Contribution:** 3 good
**Rating:** 4
**Confidence:** 4

**Summary:**

This paper considers the problem of uncertainty quantification for SVGP models with a Brownian motion prior for a non-parametric estimation problem. Theoretical results are presented that show

**Strengths:**

The results are fundamental in nature and of broad interest, and clearly explained except in places where quantities are not clearly defined.

**Weaknesses:**

The main contributions of the paper are a little unclear. It is a bit unclear which results are known and which results are new.

It is also unclear what is special about the rescaled Brownian motion prior. Does the paper recommend using the rBM prior over other priors because of some special properties. Otherwise the choice of rBM in the title is not well motivated.

Does the proof technique also extend to more general prior models beyond rBM.

The choice of rBM prior as the main focus is also not well motivated. If this is a standard choice, this must be explained clearly.

It would perhaps be useful to provide a brief sketch of the proof. For eg the discussion below Proposition 3.2 is somewhat of a proof sketch but isn't labeled as such.


**Questions:**

What precisely is Q*. Can you please define it formally in section 2. Perhaps Q* should include a gamma in the subscript. Is Q* the eigenvector SVGP or any SVGP as defined above (3).

Similarly please also provide the definition of rescaled BM in the preliminaries.

A formula for the Mattern kernel in Section 5 would be useful.

Table 1 columns currently labeled m* and n need to be renamed to something more informative and which doesnt require reading the paragraph above the table before reading the table. The caption can also be more informative (and better spaced from the table). The explicit value of m* would also be useful to know instead of the formula in the caption. A brief description of coverage, length and RMSE is necessary to read the table clearly. Alas, these are relegated to the appendix. (Similarly for Table2)

n,2 subscript in Proposition 3.4 is unclear.

Is Lemma 3.1 a noteworthy result or can it be part of section 2? Similarly the paragraph above Prop 3.2

In Section 2.1, is it GP(nu0, k) instead of m? please dont use m for mean function since m is the number of inducing points.

The space in which u_i belong is quite unclear. It appears they are actually part of the target space (ie same as space of y_i). Then the choice of the naming them Eigenvector inducing features is perhaps misleading, and Eigenvector inducing "targets" is perhaps more appropriate.

The term cov(f(x), u_i) is confusing. How is this covariance function defined? since the kernel requires two points in the domain as input. Please clarify this immediately following the definitions of Kxm etc., rather than point the reader to [38].

The bolding can be more consistent. I think r_m is a vector which is not bolded unlike other vector quantities.

In Section 3.1 end of line 150, shouldn't this also be f(x)|Dn since SVGP also is conditional on n samples?

The choice of naming tn as "frequentist variance" is unclear. Maybe a comment on that (at least in OpenReview) would be useful, but also to the reader of this paper. As suggested earlier, the discussion leading upto Prop 3.2 can be part of Section 2 since everything follows naturally from the preliminaries and does not require any special mention in the section on Main results.

The role of Mn in all results is a little unclear. An example Mn and its implications would be useful for the reader.

**Limitations:**

No limitations envisioned.

---

> ### Author Rebuttal · Authors · 2023-08-07
>
> We thank the reviewer for their constructive suggestions and helpful comments. Our response:
>
> - The main contributions of the paper are a little unclear/What is special about the rescaled Brownian motion prior?/Does the proof technique also extend to more general prior models beyond rBM?
>
> Please see our general reply.
>
> - It would perhaps be useful to provide a brief sketch of the proof.
>
> We would like to include this, but unfortunately do not have space in the main text.
>
> - What precisely is Q*?
>
> This is the SGPR based on $m$ inducing variables. This is defined in Section 2 and we have added reminders at various points in the text to help the reader, e.g. "the SGPR $Q^*$. We have also written $Q_\gamma^*$ for our main results for rescaled Brownian motion to more clearly differentiate when our results apply to general GPs, and when to rescaled Brownian motion.
>
> - Similarly please also provide the definition of rescaled BM in the preliminaries.
>
> This has been added to the start of Section 2.
>
> - Fixing Tables.
>
> We have followed your suggestions, and hopefully the table is now easier to read with a more informative caption. We have kept the definitions of the summary statistics (RMSE, etc.) in the supplement due to space constraints.
>
> - Is Lemma 3.1 a noteworthy result or can it be part of section 2?
>
> We expect that most of this is known to experts, but we have not seen it clearly written down. Given it is central to our proof, we have kept it in Section 3. In particular, the part studying the frequentist variance of the SGPR mean may not be known given there is relatively little overlap between those working on SGPR and statistical theory. While not a difficult result, we find it informative.
>
> - The space in which u_i belong is quite unclear. It appears they are actually part of the target space (ie same as space of y_i). Then the choice of the naming them Eigenvector inducing features is perhaps misleading, and Eigenvector inducing "targets" is perhaps more appropriate.
>
> Thank you for point this out. We changed this to "eigenvector inducing variables" to avoid confusion.
>
> - The term cov(f(x), u_i) is confusing. How is this covariance function defined? since the kernel requires two points in the domain as input. Please clarify this immediately following the definitions of Kxm etc., rather than point the reader to [38].
>
> We have added the definition. In our case, $u_j = \sum_{i=1}^n v_j^i f(x_i)$ and hence $cov(f(x),u_i) = \sum_{i=1}^n v_j^i k(x,x_i)$.
>
> - The bolding can be more consistent.
>
> We have tried to make this more consistent throughtout following your suggestion.
>
> - In Section 3.1 end of line 150, shouldn't this also be f(x)|Dn since SVGP also is conditional on n samples?
>
> While true, we prefer not to write the conditioning to clearly differentiate this from the full posterior, which is more traditionally written this way.
>
> - The choice of naming tn as "frequentist variance" is unclear. Maybe a comment on that (at least in OpenReview) would be useful, but also to the reader of this paper.
>
> One can consider the (sparse) posterior mean as a frequentist estimator, i.e. the randonmess comes through the data generating distribution $y \sim P_f$. Then you can study the variance of this estimator with respect to the data generating distribution. Together with the bias of the posterior mean, this measures how far this posterior centering will be on average from the truth, and hence is a key quantity to understand whether the credible intervals are correctly centered.
>
> - The role of Mn in all results is a little unclear. An example Mn and its implications would be useful for the reader.
>
> $M_n$ can be any sequence tending to infinity, arbitrarily slowly. One should think of this as 'a large enough constant' in front of the rate
> $n^{-\frac{\min(\alpha,\gamma)}{2\gamma+1}}$
> , but for technical reasons when proving posterior convergence rates, it is often easier to take this `large constant' as something tending to infinity very slowly.
>
> - All other minor points.
>
> We have corrected these subject to space.

---

### Author Rebuttal · Authors · 2023-08-07

We thank all reviewers for their constructive suggestions and helpful comments. We answer here the main issues raised by reviewers. The main changes we made are to better motivative the use of the rescaled Brownian motion prior and further discuss the connections with the Matern.

- Motivation behind using the rescaled Brownian motion (rBM) prior.

Our main motivation for using this prior is that it provides a compromise of tractable formulas allowing us to obtain precise theoretical results, while simultaneously sharing many properties with the widely used Matern GP, meaning our main qualitative conclusions also apply there. We do not necessarily advocate using rBM in practice, since the prior draws will be very rough, but our results do provide a template for what to expect when using the Matern GP in practice, which we verify is true numerically in simulations.

The tractability of rBM allows us to derive precise results, e.g. that pointwise credible sets from properly calibrated SVGPR's will be conservative (larger than optimal), and thus require no inflation as has been proposed elsewhere (e.g. Vakili et al. ICML 2022). Simulations indicate similar results holds for the Matern as predicted, but we do not know how to prove this rigorously. Since this work is, to the best of our knowledge, the first establishing exact theoretical coverage guarantees for pointwise credible sets for any SVGPR and seems to carry the right practical message for the Matern, we hope this is already an interesting message.

We expanded this motivation in our introduction and Section 4, further discussing connections with the Matern, e.g. the rescaling we consider plays the analogous role to the lengthscale parameter for the Matern.

- Extending these results to other Gaussian process priors beyond rescaled Brownian motion.

Our analysis is particular to rescaled Brownian motion since our proofs rely on explicit expressions for the eigenvectors/values for this GP. Since these are not typically available for other GPs our method of proof will not extend to these, and new ideas are required. We note that even for the true posterior, few pointwise UQ results are available, e.g. we are not aware of any exact rigorous coverage guarantees for the Matern process in nonparametric regression, let alone for sparse approximations. We first need to develop tools to deal with the true posterior before tackling the harder case of SVGPR approximations. While our proofs are indeed specific, these are the first exact pointwise inference results for SVGPRs and seem to give the correct message for the Matern prior, as we verify numerically.

- Clarify the main contributions of the paper. What results are known and which are new?

We tried to summarize our main conclusions in bullet points in the introduction. From a technical perspective, our main contributions are to derive exact asymptotic UQ guarantees and convergence rates for pointwise inference using SVGPR based on the rBM prior for different regimes depending on the smoothness of the prior and truth. As far as we know, none of our main results (theorems, propositions, etc.) are known elsewhere.

As pointed out by a reviewer, we expect Lemma 3.1 (general formulas for bias and variance of SVGPR) may be known to experts, but we haven't seen it written down exactly in this form. Given it is central to our proof, we have kept it.

- Connections with Burt et al. (ICML 2019).

Burt et al. consider approximation quality as measured by the KL-divergence between the SVGPR and true posterior. We consider the different question of whether the SVGPR behaves well for pointwise inference, even when it is not a close KL-approximation. In particular, the approximation quality may be good but the SVGPR performance will be bad if the true posterior also performs badly. Similarly, the approximation quality may not be good in KL-sense, but the SVGPR posterior can still perform well for pointwise inference.

Our results provide an example of the last situation (see Remark 3.7), where we take sufficiently few inducing points such that the KL-divergence with the true posterior increases with the sample size $n$, but the SVGPR still provides as good pointwise inference. Of course, if the true posterior performs well and the approximation is good (as in Burt et al.), then the SVGPR will also perform well. But our results show this is not necessary.

Their data-generating setup is also not directly comparable to ours. They take expectations over both the data $(x,y)$ and prior on $f$, so their results are 'averaged' over the prior. This is a more Bayesian way of thinking and such results do not translate into frequentist guarantees about convergence rates or UQ, which hold assuming the data comes from a fixed function $f_0$ (see our Assumption 2.1). This Bayesian approach allows one to ignore the method's performance for certain functions as long as they occur with small prior probability, whereas the frequentist (minimax) perspective seeks to understand how it behaves for a given (often worst-case) function. Such setups can lead to different conclusions, see e.g. Chapters 6.2-6.3 in the Bayesian nonparametric textbook of Ghosal and van der Vaart (2016).

In conclusion, we answer a fundamentally different question and work in a different setting from Burt et al, so our results are not directly comparable with theirs. We have only added some brief discussion in our manuscript due to space constraints.

Regarding frequentist results, Nieman et al. (JMLR 2021, arXiv 2022) study convergence rates and UQ for SVGPR with $L_2$ credible sets. These results crucially exploit the $L_2$ structure, which allows easier mathematical analysis, but is less reflective of practice. For pointwise inference, Vakili et al. (ICML 2022) obtain nice guarantees in a different bandit-like setting by enlarging credible intervals. In contrast, our results suggest the uninflated sets are already larger than needed.

---

### Decision · Program_Chairs · 2023-09-21

**Decision:**

Accept (poster)

**Comment:**

This paper addresses a challenging but important topic: obtaining uncertainty quantification guarantees for commonly used heuristic GP approximations. Although the rescaled Brownian motion prior is non-standard, the theoretical results nevertheless provide valuable qualitative insights.